

# Topological transitions in weakly monitored free fermions

**Graham Kells**[1,2]⋆, **Dganit Meidan**[3]† **and Alessandro Romito**[4]‡

**1** Dublin City University, School of Physical Sciences, Glasnevin, Dublin 9, Ireland
**2** Dublin Institute for Advanced Studies, School of Theoretical Physics,
Burlington Rd, Dublin 4, Ireland
**3** Department of Physics, Ben-Gurion University of the Negev, Beer-Sheva 84105, Israel
**4** Department of Physics, Lancaster University, Lancaster LA1 4YB, United Kingdom

⋆ gkells@stp.dias.ie , † dganit@bgu.ac.il , ‡ alessandro.romito@lancaster.ac.uk

## Abstract

We study a free fermion model where two sets of non-commuting non-projective measurements stabilize area-law entanglement scaling phases of distinct topological order. We show the presence of a topological phase transition that is of a different universality class than that observed in stroboscopic projective circuits. In the presence of unitary dynamics, the two topologically distinct phases are separated by a region with sub-volume scaling of the entanglement entropy. We find that this entanglement transition is well identified by a *combination* of the bipartite entanglement entropy and the topological entanglement entropy. We further show that the phase diagram is qualitatively captured by an analytically tractable non-Hermitian model obtained via post-selecting the measurement outcome. Finally we introduce a partial-post-selection continuous mapping, that uniquely associates topological indices of the non-Hermitian Hamiltonian to the distinct phases of the stochastic measurement-induced dynamics.

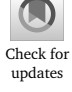

# 1  Introduction

The complex quantum dynamics of many-body systems underpins numerous fundamental physical phenomena, from the (non-)thermalization of isolated systems [1, 2] to the information scrambling in an open quantum setting [3] and chaos in black holes [4–6]. In this context, the possibility of following individual readouts in monitored quantum circuits promises a unique platform that simultaneously generates and diagnoses complex dynamical behaviours. A prominent example is the discovery that weak measurement can both induce and characterise entanglement scaling phase transitions [7–11].

The basic mechanism behind such transitions is the quantum Zeno effect, whereby frequently occurring measurements constrain the local degrees of freedom, resulting in sub-extensive entropy scaling. [7–28, 28–81]. Recent works have explored the use of this mechanism to stabilize quantum states with distinct topological order. This can be achieved via the steering of the averaged Linbladian dynamics [82] or by following the time evolution of quantum trajectories, where the dynamics is induced either by measurement alone [49] or in combination with Clifford unitaries [23, 24, 50, 61].

Many aspects of the measurement-induced topological entanglement transitions remain to be explored. One open question is the fate of the transition in the non-projective (weak) measurement setting, which is the natural one for many experimental architectures. Indeed the analysis of entanglement scaling transitions [11, 20, 32, 36–48, 65, 83–91] indicate that, due to the non-linear dependence of the back-action on the state itself, non-projective measurements induce transitions of different universality class [11, 20] than their fully projective counterparts.

Another important question deals with interpretation of topology in a statistical distribution of quantum states. So far, measurement-induced topological phase transitions have been identified via tracking average indicators, e.g. the topological entanglement entropy [92, 93], established for ground states of gapped systems. It is however unclear what non-trivial topological measures actually imply on the quantum trajectory level where generic states are effectively in the middle of the spectrum. Indeed it is not clear to what degree such measures can be reliable indicators of transitions to regimes with extensive entanglement entropy scaling.

In this work we address these questions in a free fermionic setup. Here, frequent measurements can induce a transition between area law and a critical phase with a logarithmic entanglement scaling in one-dimension [30–32, 74]. We show that competing non-projective monitoring of a free fermion system can indeed drive a topological entanglement transition between distinct area-law phases and that this transition is of a different universality class compared to projective measurement setups [23, 24, 50, 61].

With the addition of unitary dynamics, these two short-range entanglement phases are separated by an extended critical phase with a logarithmic scaling of the entanglement entropy.

To determine the transition points for this free fermion system, we introduce the combined measure of topological entanglement entropy *and* half-cut entanglement entropy that acts as an order parameter and more clearly, than either measure separately, marks the transition to critical scaling along generic cuts in the phase space. Importantly, our analysis shows that the entanglement transition in the previously studied charge conserving model [31] is *non generic*, as the indicator in this limit does not point to a transition at a finite measurement rate.

Finally, we introduce a model of partial post-selection that recovers the dynamics generated by post-selecting the measurement outcome as a continuum limit of a family of models with stochastic dynamics. This allows one to interpolate between the fully stochastic dynamics and the deterministic post-selected model, for which we can map the full phase diagram and associated topological indices. This approach provides a general way to relate topological invariants to stochastic quantum dynamics.

## 2 Model

We study a system of fermions in a one-dimensional lattice, with sites labelled by $j = 1, \ldots, L$, evolving in time under the effect of a local Hamiltonian and two sets of local continuous measurements, as sketched in Fig. 1. The corresponding dynamics is described by a stochastic Schrödinger equation (SSE) [94], which takes the form of a Wiener process for the differential evolution of the system's state, $|\psi_t\rangle$ over an infinitesimal time-step,

$$
\begin{aligned}
d\,|\psi_{t+dt}\rangle = -i\,dt \left[ H - i\frac{\gamma}{2}\sum_j \left( M_j - \langle M_j\rangle_t \right)^2 - i\frac{\alpha}{2}\sum_j \left( \tilde{M}_j - \langle \tilde{M}_j\rangle_t \right)^2 \right] |\psi\rangle_t \\
+ \left[ \sum_j \delta W_{j,t} \left( M_j - \langle M_j\rangle_t \right) + \sum_j \delta\tilde{W}_{j,t} \left( \tilde{M}_j - \langle \tilde{M}_j\rangle_t \right) \right] |\psi_t\rangle .
\end{aligned}
\tag{1}
$$

Here $H$ is the system's Hamiltonian and $M_j$ and $\tilde{M}_j$ are the Hermitian operators associated with two non-commuting positive operator valued measures (POVM) of observables at site $j$. The Wiener stochastic increments $\delta W_{j,t}$, $\delta\tilde{W}_{j,t}$ are independently Gaussian-distributed with $\langle \delta W_{j,t}\rangle = 0$, $\langle \delta W_{j,t}\delta W_{j',t}\rangle = \gamma\,dt\,\delta_{j,j'}\delta_{t,t'}$ and $\langle \delta\tilde{W}_{j,t}\rangle = 0$, $\langle \delta\tilde{W}_{j,t}\delta\tilde{W}_{j',t}\rangle = \alpha\,dt\,\delta_{j,j'}\delta_{t,t'}$. The parameters $\gamma$ and $\alpha$ control the strength of the two measurement sets, so that $1/\gamma$ and

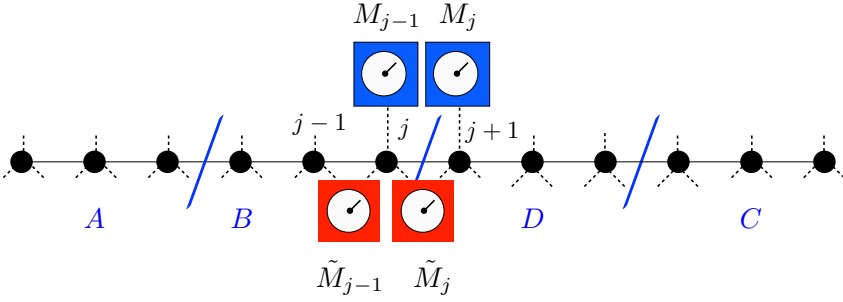

Figure 1: Monitored system and detector model. A 1-dimensional chain of fermion hopping between next-nearest sites (black dots) which are monitored by local detectors sensing the local occupation $M_j$ (upper blue boxes) and the occupation of the "Kitaev modes" $\tilde{M}_j$ (lower red boxes). Blue lines separate the regions $A, B, D$ and $C$, each of equal size $L/4$, used to define the topological entanglement entropy in Eq (4).

$1/\alpha$ set the typical time at which the system evolves close to one of the eigenstates of the measured operators, $M_j$ and $\tilde{M}_j$, respectively. Note that the presence of the expectation values $\langle M_j \rangle_t = \langle \psi_t | M_j | \psi_t \rangle$ make the evolution a non-linear function of the state $|\psi_t\rangle$. It is useful in the following to regard Eq. 1 as the result of the detection of $M_j$ via a quantum pointer of coordinate $x_j$ linearly coupled to the observable $M_j$ so that the pointer's readout is given by $x_j = \gamma dt \langle \psi_t | M_j | \psi_t \rangle + \delta W_{j,t}$, as reported in Sec. 2.1 and detailed in Appendix A. Similarly for the detection of $\tilde{M}_j$, $\tilde{x}_j = \alpha dt \langle \psi_t | \tilde{M}_j | \psi_t \rangle + \delta \tilde{W}_{j,t}$.

Hereafter, we specify the Hamiltonian to describe nearest neighbour hopping,

$$H = w \sum_{j=1}^{L-1} c_{j+1}^\dagger c_j + \text{h.c.}, \tag{2}$$

with $c_j$ the fermionic annihilation operators at site $j$, and $w$ the magnitude of the hopping measurement, and the measurement to two non-commuting sets of observable:

$$
\begin{aligned}
M_j &= 2c_j^\dagger c_j - 1, \\
\tilde{M}_j &= 2d_j^\dagger d_j - 1 = \left( c_{j+1}^\dagger - c_{j+1} \right)\left( c_j^\dagger + c_j \right),
\end{aligned} \tag{3}
$$

where $d_j = (c_j + c_{j+1} + c_j^\dagger - c_{j+1}^\dagger)/2$ with $0 \le j \le L-1$.[1] Eq. (2) assumes open boundary conditions, but can be generalised to periodic boundary conditions with $L \to L+1$ and the identification $L+1 = 1$. The operator $M_j$ is the occupancy of site $j$ (relative to half-filling) and requires a detector coupling to a single chain site, while the back-action from $\tilde{M}_j$ requires detectors that couple to pairs of adjacent sites (cf. Fig. 1). The physical meaning of the operator $\tilde{M}_j$ can be understood by considering that the modes $d_j$ are the eigenmodes of the Kitaev Hamiltonian for spinless p-wave superconductor [95] in the limit of single-site coherence length, $H_K = \alpha \sum_{j=1}^{L} \left( c_j^\dagger c_{j+1} + c_j^\dagger c_{j+1}^\dagger \right) + \text{h.c.} = 2\alpha \sum_{j=1}^{L-1} \left( d_j^\dagger d_j - 1/2 \right)$. The operator $\tilde{M}_j$ corresponds to the occupation of those modes (relative to half-filling). A possible measurement scheme for the detection of $\tilde{M}_j$ is presented in Appendix A.

The models displays competition between the free unitary dynamics, which generates extensive entanglement, and the local density measurements, which drive the system towards a disentangled state with a well defined site occupancy. The resulting disentanglement transition has been studied for free fermionic systems for single on-site density measurements [31, 32, 48].

In this work, the additional "Kitaev-density", $\tilde{M}_j$, preserves the Gaussian nature of the states but drives the system towards a distinct disentangled state characterised by a definite occupancy of the $d$-modes. Crucially this term has introduced competition between two non-commuting measurements, each trying to localise the system onto different short-range entangled states. Importantly, when this additional $\tilde{M}_j$ measurement term dominates, the trajectories flows to states with clearly identifiable symmetry protected topological order.

In order to study the entanglement and topological properties of the system state, we will use a combination of the topological entanglement entropy [61, 92, 93, 96, 97] and half-cut entanglement entropy. The topological entanglement entropy is devised to be a generic indicator of topological ground states of gapped Hamiltonian systems and it is defined as

$$\bar{S}_L^{\text{top}} = \bar{S}_L^{AB} + \bar{S}_L^{BC} - \bar{S}_L^{B} - \bar{S}_L^{ABC}, \tag{4}$$

where $S_L^X = -\text{Tr}[\rho_X \ln \rho_X]$ is the Von Neumann entropy computed for the reduced density matrix $\rho_X$ associated with the region $X$, and defined for the specific cuts configurations in Fig.

---

[1]Note that, under the Jordan-Wigner transformation, the measurements of $M_j$ and $\tilde{M}_j$ are mapped to measurements of $S_{z,j}$ and $S_{x,j}S_{x,j+1}$, respectively.

1(a).[2] The half cut entanglement entropy is the entanglement entropy where the partial trace is taken over precisely half of the wire $\bar{S}_L^{AB} = \bar{S}_L^{CD}$ and will denoted in shorthand $\bar{S}_L \equiv \bar{S}_L^{AB}$ in what follows. Here $\bar{\cdot}$ denotes average over stochastic fluctuations.

The topological entanglement entropy has been employed to diagnose a measurement induced topological phase transition from stroboscopic projective measurements in a stabiliser circuit model [23]. Crucially, the entanglement entropy is a non-linear function of the system's density matrix, so its average depends on the full distributions of states rather than the average state of the system described by the averaged density matrix $\bar{\rho}$. Due to the Gaussian nature of the states of the model, the half-cut and topological entanglement entropy can be computed from 2-points correlation functions $C_{ij}(t) = \langle \psi(t)| c_i^\dagger c_j |\psi(t)\rangle$ and $F_{ij} = \langle \psi(t)| c_i c_j |\psi(t)\rangle$ [41, 98, 99] as described in Sec. 2.1 section along with the procedure for the numerical simulations.

## 2.1 Implementation of the time-evolution

We simulate numerically the time evolution of a generic initial state in Eq. (1) via quantum monte-carlo where the time evolution is determined over sufficiently small time-steps. Each of the measurement of the operators $M_j$ can be written as $M_j = a_+ \Pi_{j,+} + a_- \Pi_{j,-}$ in terms of projectors into its eigenstates $\Pi_{j,\pm}$ and corresponding eigenvalues $a_\pm = \pm 1$. We consider the response of a one-dimensional pointer acting as a detector linearly coupled to $M_j$ [94, 100]. The associated Kraus operator is $K_j(x, \lambda) = \sqrt{G(x + a_+)} \Pi_{j,+} + \sqrt{G(x + a_-)} \Pi_{j,-}$, where $G(x) = \exp(-x^2/2\lambda^2)/\sqrt{2\pi}\lambda$ and the parameter $\lambda^2/dt = \gamma$ quantifies the measurement backaction. For a state $|\psi_t\rangle$ the measurement outcome for the $j$-th operator is drawn from the distribution

$$P_j(x) = \langle \psi_t| K_j^\dagger(x, \lambda) K_j(x, \lambda) |\psi_t\rangle \,, \tag{5}$$

and, for a given $x$, the state evolves to

$$|\psi_t\rangle \to \mathcal{L}_\gamma(|\psi\rangle) = \prod_j K_j(x, \lambda) |\psi_t\rangle / \mathcal{N}_\gamma \,, \tag{6}$$

where, in the limit of small $dt$, the infinitesimal state evolution reduces to $\mathcal{L}_\gamma(|\psi_t\rangle) = (1/\mathcal{N}_\gamma) e^{\sum_j (\gamma \langle M_j \rangle dt + \delta W_j) M_j} |\psi\rangle$, where $\mathcal{N}_\gamma$ is the proper state normalization. This coincides with the SSE in Eq. (1) at $\alpha = 0$, after noting that $M_j^2 = 1$. In the presence of two non-commuting measurements and hopping Hamiltonian we can write

$$|\psi_{t+dt}\rangle = (1/\mathcal{N}) e^{\sum_j (\gamma \langle M_j \rangle dt + \delta W_j) M_j + \sum_j (\alpha \langle \tilde{M}_j \rangle dt + \delta \tilde{W}_j) \tilde{M}_j} e^{-iHdt} |\psi\rangle = \mathcal{L}_\gamma \left( \mathcal{L}_\alpha \left( e^{-iHdt} |\psi_t\rangle \right) \right) |\psi_t\rangle \,, \tag{7}$$

where we have defined $\mathcal{L}_\alpha(|\psi\rangle) = (1/\mathcal{N}_\alpha) e^{\sum_j (\langle \tilde{M}_j \rangle \alpha dt + \delta \tilde{W}_{j,t}) \tilde{M}_j} |\psi_t\rangle$. Note that, in the last equality, the effect over an infinitesimal step is obtained by a trotterization of the evolution operator and corresponds to the form used in the numerical implementation.

Since we are typically interested in the long-time dynamics of the system, the result does not generally depend on the choice of the initial state - although consideration of symmetries is sometimes required. For $\alpha = 0$ our system is particle-number conserving and thus steady states will have the same number of particles as the initial state. For non-zero $\alpha$, only fermionic parity is preserved. For all simulations we choose an initial half-filled state with fermions occupying the odd sites of the chain only.

The simulation of the stochastic process allows us to obtain the time-evolution of the full probability distribution of pure states of the system as opposed to the evolution of the averaged

---

[2]While the topological entanglement entropy can be efficiently computed in free fermionic systems, the possibility of tracking $\bar{S}_L^{\text{top}}$ experimentally poses the same challenges as detecting $\bar{S}_L$. Ultimately this comes down to extracting information about the entanglement of a system's state along post-selected trajectories. Some early experimental advances in this respect have been recently reported [76, 119].

density matrix of the system. The former is essential to provide access to the entanglement properties of the system and its scaling, to which the average evolution is instead oblivious.

Practically speaking, as all operators are quadratic in the number of fermions but not necessarily number conserving, we implement this calculation within the Bogoliubov de Gennes (BdG) formalism. We represent our states as the

$$|\psi(t)\rangle = \prod_{n=1}^{L} \beta_n(t) |0\rangle \,, \tag{8}$$

where the $\beta^\dagger$ and $\beta$ operators are encoded as a $2L \times 2L$ matrix of orthonormal vectors

$$\mathsf{W}(t) = \begin{pmatrix} U(t) & V(t)^* \\ V(t) & U(t)^* \end{pmatrix} \,, \tag{9}$$

such that $\beta_n^\dagger = \sum_x U_{x,n} c_x^\dagger + V_{x,n} c_x$ and $\beta_n = \sum_x V_{x,n}^* c_x^\dagger + U_{x,n}^* c_x$ and $|0\rangle$ is the state with all $c$-fermion sites of the chain left unoccupied. We iterate the state forward in time by updating the first $N$ columns of W according to

$$\begin{pmatrix} U(t+\delta t) \\ V(t+\delta t) \end{pmatrix} = O\left( e^{\mathsf{L}_\alpha \delta t} O\left( e^{\mathsf{L}_\gamma \delta t} e^{-i\mathsf{H}\delta t} \begin{pmatrix} U(t) \\ V(t) \end{pmatrix} \right) \right) \,, \tag{10}$$

where H , $\mathsf{L}_\gamma$, and $\mathsf{L}_\alpha$ are $2N \times 2N$ matrices encoding the quadratic Hamiltonian and measurement operators in BdG form. The operations $O$ represent an orthonormalisation step of the $2N \times N$ matrix implemented via Gram-Schmidt or singular value decomposition.

The representation can be used to conveniently render overlaps between different states

$$|\langle \psi_1 | \psi_2 \rangle| = \left| \sqrt{\det(U_1^\dagger U_2 + V_1^\dagger V_2)} \right| \,, \tag{11}$$

but more importantly for our purposes it also allows us to efficiently calculate single particle correlators

$$\mathsf{C}_{i,j}(t) = \begin{pmatrix} C_{ij}(t) & F_{ij}(t) \\ -F_{ij}^*(t) & \delta_{ij} - C_{ij}(t) \end{pmatrix} \,, \tag{12}$$

here $C_{ij}(t) = \langle \psi(t)| c_i^\dagger c_j |\psi(t)\rangle$ and $F_{ij} = \langle \psi(t)| c_i c_j |\psi(t)\rangle$ as $C = V^* V^T$ and $F = V^* U^T$, from which we can directly calculate the entanglement entropy, see e.g. [41, 98, 99] via $S_X = -\text{Tr}[\mathsf{C}_X \ln \mathsf{C}_X + (1 - \mathsf{C}_X)\ln(1 - \mathsf{C}_X)]$.

## 3 Results

In this section we discuss our numerical results based on topological entanglement entropy and half cut entanglement entropy. We first summarize our main finding.

We initially focus on the dynamical evolution guided by the two competing weak measurements. We find that the system transitions between two area law phases which differ by the topological entanglement entropy. The critical behavior of this entanglement transition differs from the one obtained in a projective measurement setting, which indicates that dynamical evolution generated by the two models is of a *different universality class*.

We then study the full evolution including unitary dynamics. Here the two area law phases are separated by an extended critical phase with logarithmic entanglement scaling. We argue that, because the topological entanglement entropy saturates for large system sizes, it is not sufficient by itself to act as an order parameter. To this end we show that in free fermionic

systems the combination of $\bar{S}^{\text{top}}$ and $\bar{S}_L$ is needed to clearly mark the entanglement transition across generic cuts of the phase diagram. Interestingly, our analysis shows that the transition in the previously studied charge conserving model [31] is non generic, where our numerical data does not point to a transition at finite measurement rate.

Lastly we consider the deterministic evolution obtained by post selecting the measurement outcome. This model exhibits a qualitatively similar dynamical phase diagram which we can analyze and assign topological indices. We then introduce a model of partial post-selection which links the full stochastic evolution with the post selected dynamics and present numerical results on the relation between the two limiting behaviors.

## 3.1 Measurement-only induced topological transition

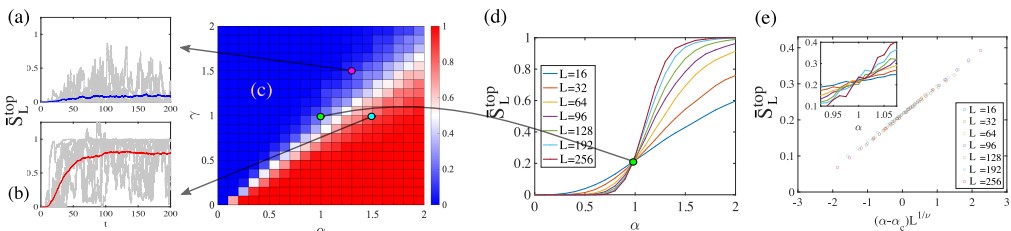

Figure 2: (a) and (b) Topological entanglement entropy for measurement-only dynamics ($w = 0$). (c) $\bar{S}^{\text{top}}$ as a function of the two measurement strengths ($\alpha, \gamma$). The topological entanglement entropy distinguishes the topologically trivial phases ($\bar{S}_L^{\text{top}} = 0$) from the topologically non-trivial one ($\bar{S}_L^{\text{top}} = 1$). The right panels report typical time traces of $S_L^{\text{top}}$ (grey curves) and corresponding averages $\bar{S}_L^{\text{top}}$ (red and blue curves) in the two regions. (d) $\bar{S}_L^{\text{top}}$ as a function of $\alpha$ at constant $\gamma = 1$ for different system sizes $L$. The crossing point between $\bar{S}_L^{\text{top}}$ between subsequent system sizes is used to identify the critical transition point [green dot in panel (a)]. (e) Rescaled linear fit of the data in (b) close to the transition point $\alpha = 1$ reveals best fit data collapse with $\alpha_{cr} = 0.9(9)$ and $\nu = 1.6(7)$. The results are obtained for an initial state at half filling with alternating occupancies of the chain sites.

We consider first the measurement-only dynamics generated by the two competing non-commuting measurements by setting $w = 0$ in Eq. 2. While each of the two measurements seeks to drive the system to disentangled states associated with short range entanglement, the two phases are distinguished by the different topological properties of their steady states. This is evident by examining the statistical distribution of steady states in the limiting cases $\gamma = 0$ and $\alpha = 0$. At $\alpha = 0$, for an initial state of filling fraction $n$, any state of the form $|\psi\rangle_C = \prod_i p_i c_i^\dagger |0\rangle$ with $p_i = 0,1$ and $\sum_i p_i = n$ is a fixed point of the evolution. Similarly, in the case of $\gamma = 0$, the steady state will be a drawn from a statistical distributions of states of the form $|\psi\rangle_K = \prod_i p_i d_i^\dagger |0'\rangle$, where $|0'\rangle$ is now the state annihilated by the $d$ operator. While both configurations correspond to localised short-range entanglement states, the latter has a topologically protected $\mathbb{Z}_2$ degeneracy for open boundary conditions since all the states of the form $|\psi\rangle_K$ are eigenstates of the topologically non trivial Hamiltonian $\tilde{H} = \alpha \sum_{i=1}^L \left( c_i^\dagger c_{i+1} + c_i^\dagger c_{i+1}^\dagger \right) + \text{h.c.}$.

The difference between these two limiting behaviors is reflected in time traces of the topological entanglement entropy, reported in Fig. 2(a) for $\alpha \ll \gamma$ and Fig. 2(b) for $\alpha \gg \gamma$. After an initial transient period, $S_L^{\text{top}}$ fluctuates around $\bar{S}_L^{\text{top}} = 0$ or $\bar{S}_L^{\text{top}} = 1$, respectively. Notably, while the trajectory-averaged entanglement entropy reaches a well defined steady state, the (topological) entanglement entropy for the individual trajectories fluctuates even in the steady

state regime. This is due to the competition between the two measurements. For example, for $\alpha \gg \gamma$, while the dominant set of measurement tends to stabilise a state of the form $|\psi\rangle_K$, repeated applications of $M_i$ measurements induce transitions between different such eigenstates (all with $S^{\mathrm{top}} = 1$). Since these "jumps" occur at random times, they are averaged out in $\bar{S}_L^{\mathrm{top}}$. It is also worth noting that the numerical simulations confirm that the stationary average half-cut entanglement, $\bar{S}_L$, does not depend on $L$, corresponding to an area law entangled state.

From the average steady state topological entanglement entropy $\bar{S}_L^{\mathrm{top}}$, we obtain the phase diagram for the two measurement model with $w = 0$ as shown in Fig. 2(c) for a system of size $L = 96$ with open boundary conditions. The topological entanglement entropy clearly shows the trivial and non-trivial topological phases with $\bar{S}_L^{\mathrm{top}} = 0$ and $\bar{S}_L^{\mathrm{top}} = 1$ respectively and a smooth crossover between the two around $\gamma \approx \alpha$. A sharp phase transition is expected in the thermodynamics limit, with two distinct area-law entangled phases characterised by $\lim_{L \to \infty} \bar{S}_L^{\mathrm{top}} \to 0$ and $\lim_{L \to \infty} \bar{S}_L^{\mathrm{top}} \to 1$. The transition line can be determined exactly in this instance noting that (i) after a time rescaling $t \to \gamma t$, Eq. (1) is controlled by a the single parameter $\alpha/\gamma$, and (ii) Eq. (1) is invariant under the duality transformation

$$c_j \leftrightarrow d_j, \, \alpha \leftrightarrow \gamma, \tag{13}$$

where we have set $d_L = (c_L + c_1 + c_L^\dagger - c_1^\dagger)/2$. Consequently, the duality fixes the phase transition at $\alpha = \gamma$. We confirm this numerically in Figure 2 (calculating the critical point at $\alpha_{cr} = 0.9(9)\gamma$) by analysing the crossing point of $\bar{S}_L^{\mathrm{top}}$ for different system sizes $L$.

The numerical analysis allows further characterisation of the universality of the transition by evaluating its finite size scaling universal exponent. For this goal, the critical value $\alpha_{cr} = \gamma$ is used to rescale the data in the vicinity of the transition point according to

$$\bar{S}^{\mathrm{top}}(\alpha, L) = F((\alpha - \alpha_c)L^{1/\nu}), \tag{14}$$

where $\nu$ is used as a fitting parameter. The inset of Fig. 2(e) shows the raw data in the vicinity of the transition for different $L$. Rescaling of the linear fits to the raw data is shown in Fig. 2(d), where the data collapse is achieved with $\nu = 1.6(7) \approx 5/3$. The value of $\nu$ differs from the scaling of a topological phase transition from projective measurements analogues of the model [23,49,53,62,101], which can be mapped onto a classical 2D percolation model with $\nu = 4/3$ universal exponent. This result shows that weak continuous-time measurements drives the critical point toward a different universality class.

Accounts of a different universal behavior of entanglement transitions in weak vs. projective measurements have also been reported for interacting models [11,20]. Indeed, the projective nature of the measurement is essential to the mapping of stroboscopic projective models to classical percolation, and the mapping cannot be directly applied to generic weak measurements. This is because unlike projective models (i) the back action of weak measurements on the state results in a nonlinear change that depends on the state of the system itself, and (ii) in the continuum limit, the time step and the weak measurement strength both go to zero such that their ratio remains fixed. Importantly, weak measurements do not disentangle the measured degree of freedom but rather only weaken the entanglement to a degree set by the state itself, which hinders the mapping to percolation.

Finally we remark on the topological nature of the transition. We note that while the two short range entanglement phases are distinguished by their average topological entanglement entropy, no topological indices or exact degeneracy of the spectrum can be associated to the individual realizations due to their fluctuations [cf. Fig. 2(a) and (b)]. Similarly, the topological protection characteristic of ground states of gapped systems is not directly applicable to the measurement induced dynamics, which deals with steady states at arbitrary energies. Thus the notion of topological order in stochastic quantum dynamics requires further refinement.

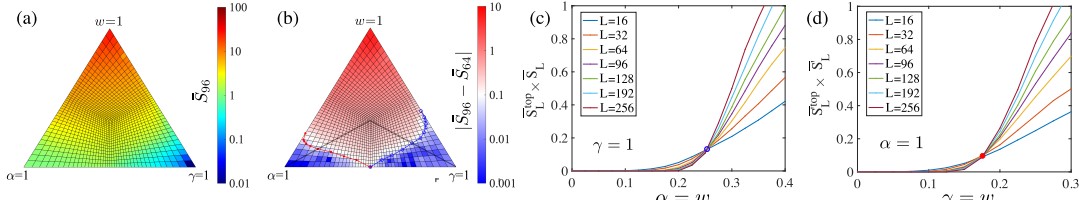

Figure 3: Phase diagram of full model (a) Density plot of $\bar{S}_L$ as a function of $w$, $\gamma$ and $\alpha$ ($w+\gamma+\alpha = 1$) in barycentric coordinates for $L = 96$. Area law scaling is recovered in the limits $\alpha = 1$ and $\gamma = 1$ with $\bar{S}_{96} = 1$ and $\bar{S}_{96} = 0$ respectively, while $w = 1$ displays volume-law scaling. (b) $\bar{S}_{96}-\bar{S}_{64}$. The entanglement entropy difference how a sharp crossover between the three distinct phases. Transition points between the sub-volume-scaling and trivial area-law scaling (blue and red circles) and between the sub-volume-scaling and topological area-law scaling are determined by crossings in the measure $\bar{S}_L^{\text{top}} \times \bar{S}_L$ (c) Data showing $\bar{S}_L^{\text{top}} \times \bar{S}_L$ ($c$−fermion basis) for fixed $\gamma = 1$ and $\alpha = w$. (d) Data showing $\bar{S'}_L^{\text{top}} \times \bar{S'}_L$ ($d$-fermion basis) according to (13), for fixed $\alpha = 1$ and $\gamma = w$. In the barycentric coordinates these two data sets fall along the dotted lines shown in panel (b).

## 3.2 Dynamics under unitary evolution and competing measurements

In the presence of unitary dynamics, i.e. $w \neq 0$, the two short range entanglement phases are separated by a critical-scaling phase with long range entanglement scaling like $\log L$. This is illustrated in Fig. 3 (a), where the limiting cases of trivial area-law scaling ($\gamma = 1$, $\alpha = w = 0$), topological area-law scaling ($\alpha = 1$, $\gamma = w = 0$), and critical scaling ($w = 1$, $\gamma = \alpha = 0$) are indicated by $\bar{S}_L \approx 0$, $\bar{S}_L \approx 1$ and $\bar{S}_L \gg 1$, respectively.

The appearance of a critical scaling phase is expected for dynamics induced by a free fermion Hamiltonian and local density measurements [30, 31]. However, determining the critical phase boundary based on the half cut entanglement $\bar{S}_L$ is inconclusive [31, 102], since it is numerically difficult discern a constant function from a logarithmic scaling with a small pre-factor. Moreover, the topological entanglement entropy $\bar{S}_L^{\text{top}}$ saturates at large system sizes in the critical phase. To overcome this difficulty we use combination $B_L = \bar{S}_L^{\text{top}} \bar{S}_L$ to discriminate between the different phases. This allows us to identify the phase boundary as a crossing point for finite size systems, which can be extrapolated to the thermodynamic limit. In the critical scaling region, $\bar{S}_L \propto \log L$ while $\bar{S}_L^{\text{top}} \propto \mathcal{O}(1)$, leading to an extensive $B_L \xrightarrow{L\to\infty} \infty$; Conversely in the topologically trivial phase, $\bar{S}_L^{\text{top}} \xrightarrow{L\to\infty} 0$ and $\bar{S}_L \propto \mathcal{O}(1)$, resulting in a vanishing $B_L \xrightarrow{L\to\infty} 0$, as illustrated in Fig. 3(c).

The corresponding critical values are marked in blue dots in panel (b), where the density plot showing difference between the entanglement half cuts obtained for different system sizes $\bar{S}_{96} - \bar{S}_{64}$ gives a qualitative indication of the different phases. (Regions of phase space associated with short range entanglement with weak dependence on system size $L$ would thus appear as dark blue.)

In order to identify the phase transition between the topologically non-trivial short range phase and the critical scaling one, we take advantage of the duality transformation in (13) and construct the analog of $B_L$ treating the $d_i$ operators as the physical fermions of the system.[3] Specifically, this can be achieved by studying the lattice model in the $d$-fermion basis. In this basis the unitary dynamics in Eq. (1) is rewritten as $H' = \frac{w}{2} \sum_i d_{i+1}^\dagger d_{i-1} + d_{i+1}^\dagger d_{i-1}^\dagger + 2d_i^\dagger d_i - 1 + \text{h.c.}$

---

[3]In terms of the local Majorana operators which makeup the fermionic degrees of freedom, this is akin to shifting the unit cell.

In particular, in this basis, $\tilde{M}_j$ measures local density. Regarding this as a dual system where $d_j$, $d_j^\dagger$ play the role of physical fermions, we introduce a dual entanglement entropy as in Eq. (12) through the correlators $C'_{ij}(t) = \langle\psi(t)| d_i^\dagger d_j |\psi(t)\rangle$ and $F'_{ij}(t) = \langle\psi(t)| d_i d_j |\psi(t)\rangle$

$$S'_X = -\text{Tr}\left[ C'_X \ln C'_X + (1 - C'_X)\ln(1 - C'_X) \right].$$

(15)

The resulting dual topological entanglement entropy $S'^{\text{top}}_L$ vanishes in the thermodynamic limit $S'^{\text{top}}_L \xrightarrow{L\to\infty} 0$ in the non-trivial topological phase $\left(\text{and } S'^{\text{top}}_L \xrightarrow{L\to\infty} 1 \text{ in the topologically trivial one}\right)$. As a consequence, we can use $B'_L \equiv \bar{S}'^{\text{top}}_L \bar{S}'_L$ to discriminate the topologically non-trivial phase form the critical scaling one. The crossing of $B'_L$ for different system sizes are shown in Fig. 3(d) and the critical values obtained are marked in red dots in panel (b).

The resulting phase diagram is qualitatively similar to that of circuits with projective measurements and interacting unitary dynamics [61], where the role of the volume law scaling phase is replaced by the critical logarithmic scaling phase. However, unlike the volume law scaling in interacting circuits, the stability of the critical phase in the thermodynamic limit of free fermionic circuits is not easily established, and it has been questioned altogether in some models [102]. Indeed, our newly introduced marker for the area-to-critical scaling transition, $B_L$, *does not* point to a transition at finite measurement rate when applied to the particle conserving dynamics $\alpha = 0$, previously studied in [31], as shown in Fig. 3 (cf. Appendix B for further analysis).

Conversely, we show that the critical scaling phase is stabilized by the presence of a second measurement. Moreover, we observe that even for dynamical evolution constraint by a single density measurement, the critical scaling phase is generically present, as can be deduced from the limit $\gamma = 0$. (Here, after the duality map, the system evolves under local density monitoring, but with the particle non-conserving Hamiltonian $H'$). Fig. 3 shows a clear finite measurement rate for the transition.

### 3.3 From stochastic evolution to non-Hermitian dynamics

Our final aim is to analyze the stochastic evolution that gives rise to an entanglement transition of a new universality class, and the corresponding dynamical phase diagram. To make progress, we consider a related deterministic evolution generated by post-selecting the measurement readouts. We show that under post-selection, the system undergoes an entanglement transition driven by the competing measurement rates. The resulting phase diagram is qualitatively similar to the stochastic dynamics and characterized by two short range entanglement scaling phases, which correspond to topologically distinct ground states of the effective non-Hermitian system, separated by a gapless phase with critical entanglement scaling. Next we show that the post selected dynamics and the stochastic evolution generated by the random measurement outcome are both limiting behaviors of one parent model with partial post selection. This partial post selected model establishes a paradigm to study the relation between these two limiting behaviors.

#### 3.3.1 Post-selected dynamics

The topological transition in the model originates from the competing stationary states stabilised by the Zeno effects of the two non-commuting measurements [50]. In order to capture this effect, it is possible to analyse sequences of predetermined measurements readouts, that stabilises one of the possible measurement eigenstates, i.e. a post-selected dynamics, which has been shown to capture entanglement transitions [41, 44, 48].

The process can be readily described in terms of detector outcome $x_j = \gamma dt\langle\psi_t|M_j|\psi_t\rangle + \delta W_{j,t}$. For the case of a the local density measurement, for example, it would correspond to selecting

the outcome $x_j = 1$. The result is a deterministic evolution of the system via a non-Hermitian Hamiltonian (cf. Sec. 2)

$$\mathcal{H} = H_0 - i\gamma \sum_{i=1}^{L} M_i - i\alpha \sum_{i=1}^{L-1} \tilde{M}_i\,, \tag{16}$$

where $H_0$ is the Hermitian Hamiltonian in (2) and the measurements operators are defined in Eq. (3). The corresponding post-selected dynamics can be studied analytically by considering the time evolution of the quantum state $|\psi(t)\rangle = \frac{\mathcal{U}(t)}{\sqrt{Z}}|\psi(t=0)\rangle$ governed by the non-unitary evolution operator $\mathcal{U}(t) = \prod_{dt=1}^{\text{Int}[t/dt]} \exp(-idt\mathcal{H})$ followed by a normalization of the wave function $Z = \langle\psi(t)|\psi(t)\rangle$ [41].

Under the evolution $\mathcal{U}(t)$ the state $|\psi(t)\rangle$ remains Gaussian at all times. Consequently the entanglement is entirely expressed through the correlation matrix (12). We derive and solve the equations of motions for the correlation matrix $C_{i,j}(t)$ [41, 98, 99], see Appendix C.

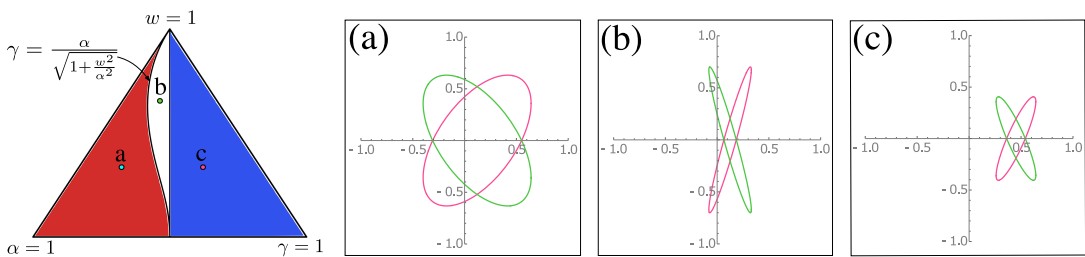

Figure 4: The phase diagram of the effective non Hermitian model. Here blue marks a gapped topologically trivial phase with $\nu_{1,2} = 0$, red marks a gapped topologically non trivial regime $\nu_{1,2} = 1$ and white marks a gapless phase. The insets (a)-(c) show the winding of $q_1(k)$ (green curve) and $q_2(k)$ (red curve) around the origin in the gapless, topological and trivial phases, respectively.

Under the post selected dynamics, the quantum state evolves to the dark state of the effective Hamiltonian (16) $\mathcal{H} = -i\mathcal{K}$, which, in turn, is identified with the ground state of $\mathcal{K}$. Explicitly, $\mathcal{K} = \sum_k \Psi_k^\dagger \mathcal{K}_{BdG}(k)\Psi_k$, where $\Psi_k = (c_k, c_{-k}^\dagger)^T$ and the corresponding BdG Hamiltonian is:

$$\mathcal{K}_{BdG}(k) = (\xi(k) + iw(k))\tau_z + \Delta(k)\tau_y\,, \tag{17}$$

where $\xi(k) = \alpha \cos k + \gamma$, $\Delta(k) = \alpha \sin k$ and $w(k) = w \cos k$.

Eq. (17) shows that $\mathcal{K}_{BdG}(k)$ coincides with the Kitaev chain with non Hermitian hopping. Besides charge conjugation symmetry, this model possesses a chiral symmetry which can be made apparent by rotating to the chiral basis by the unitary transformation $\mathcal{R} = i\tau_y$:

$$\mathcal{K}_{BdG}^{\text{chiral}}(k) = \begin{pmatrix} 0 & q_1(k) \\ q_2(k) & 0 \end{pmatrix}, \tag{18}$$

where

$$q_{1,2}(k) = \xi(k) + i[w(k) \pm \Delta(k)]\,. \tag{19}$$

In the Hermitian limit $w = 0$, $q_1(k) = q_2(k)^*$ and the complex vector $q_1(k) = \gamma + \alpha e^{-ik}$ marks a circle of radius $\alpha$ centred around $\gamma$ and the two gapped phases are discriminated by whether the resulting circle encloses the origin, with a topological phase transition at $\alpha = \gamma$. In the presence of unitary dynamics $w \neq 0$, the two complex vectors mark two tilted ellipses.

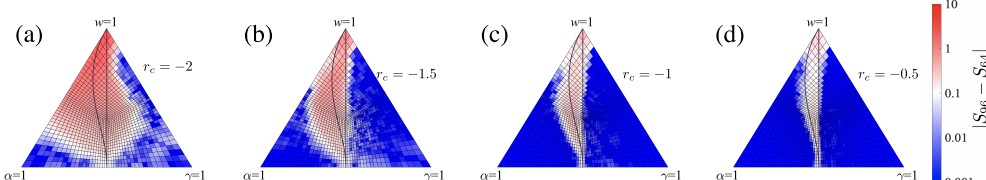

Figure 5: Density plot of $|\bar{S}_{96} - \bar{S}_{64}|$ in the partially post-selected model in Eq. (20) with (a) $r_c = -2$, (b) $r_c = -1.5$, (c) $r_c = -1$, (d) $r_c = -0.5$. Full black line correspond to the phase boundaries obtained from non-Hermitian dynamics (cf. Eq. (17)). The distinct phases of trivial area-law, topological area-law, and volume-law scaling are continuously deformed from the full measurement limit (a) to the strongly post-selected limit (b). The phase boundaries from the sharp crossover of $\bar{S}_{96} - \bar{S}_{64}$ agree well with the analytical predictions in the limit of strong post-selection [cf. panel (c)].

Whether these ellipses encapsulate the origin determine the topological phases as shown in Fig. 4 – see Appendix C for a full analysis of the model and its properties.

The entanglement properties of the post-selected quantum state are determined by the corresponding dark phases of the effective non-Hermitian model. The transition lines between the two short range entangled states and the critical scaling phase are marked as solid black line in Fig. 5, where they can be compared with the transitions obtained from the numerical analysis of the full stochastic dynamics.

The post-selected dynamics captures some of the key qualitative features of the stochastic dynamics. In particular, the presence of three distinct phases corresponding to topologically trivial and non-trivial short entanglement phases as well as the presence of a finite size region characterised by critical scaling $\propto \log L$ of $\bar{S}_{L/2}$. Yet, some features are missed by the post-selected evolution. In particular the stochastic dynamics exhibits a transition from an area-law to a critical sub-volume-law scaling of the entanglement entropy at a finite critical value of $\alpha/w$ in the limit of $\gamma = 0$. The post-selected dynamics, instead, produces area-law scaling for any $\alpha \neq 0$ at $\gamma = 0$. Similarly we note that the post selected evolution produces area-law scaling for any measurement rate $\gamma$ along the particle conserving line $\alpha = 0$ [31].

### 3.3.2 Interpolation between post-selected and stochastic dynamics

The qualitative similarity between the stochastic and post-selected dynamics can be understood by continuously interpolating between the two regimes. To achieve this, we take advantage of our continuous readout detector and introduce a new *partial-post-selection* measurement scheme where we include the measurement-induced stochastic fluctuations with a gradual weight beyond the deterministic non-Hermitian dynamics. The idea is retain only a subset of the readout data.

For clarity, we discuss this procedure for the case of $\alpha = 0$ below. The generic case for $\alpha \neq 0$ readily follows. We consider the detector's readout $x_j = \gamma dt \langle \psi_t | M_j | \psi_t \rangle + \delta W_{j,t}$, with its corresponding probability distribution, $P(x)$, as discussed in Sec. 2.1. We modify the measurement procedure by post-selecting the measurement outcome $x$, so that we retain only values such that $x \geq r_c$ and discard any value for which $x < r_c$, where $r_c \in \mathbb{R}$ is a parameter that controls the degree of our post-selection. This procedure amounts to replacing the probability distribution (5) with

$$P_{r_c}(x) = \begin{cases} 0 & \text{if } x < r_c \\ P(x) & \text{if } x > r_c \end{cases}. \tag{20}$$

For $r_c \ll -1$, $P_{r_c}(x) \approx P(x)$ and the model reproduces the fully stochastic (non post-selected dynamics). In the opposite limit, $r_c \gg 1$, as long as the readout $x$ is retained (post-selected), the corresponding dynamics is controlled by small fluctuation on top of the deterministic evolution dictated by the non-Hermitian Hamiltonian $\mathcal{H}_\gamma = -i\gamma \sum_{j=1}^{L} M_j$.

For intermediate values of $r_c$ the model continuously bridges between fully stochastic and fully post-selected dynamics. The results are presented in Fig. 5 for several values of $r_c$. The phase diagram agrees well with the post-selected non-Hermitian dynamics already for $r_c < 0$ [cf. Fig. 5(c)] for which the update of $\langle \Pi_{+,j} \rangle$ is exponentially suppressed compared to $\langle \Pi_{-,j} \rangle$, having introduced $\Pi_{j,+} = c_j^\dagger c_j$ and $\Pi_{j,-} = 1 - c_j^\dagger c_j$ as the projectors onto the occupied and unoccupied $j$-th site. For $r_c < 0$, as the bias between $\langle \Pi_{+,j} \rangle$ and $\langle \Pi_{-,j} \rangle$ from the stochastic infinitesimal update starts to fade off, the approximation in terms of the post-selected dynamics breaks down. Yet, the phase diagram continuously interpolates between that induced by post-selected evolution to the fully stochastic dynamics as we change $r_c$. In particular, already at finite values of $r_c$ the steady state exhibits a transition between an area-law and a critical scaling region at a *finite* critical value of $\alpha/w$ for $\gamma = 0$, see Fig. 5 (d). The evolution of the transition along the particle conserving line $\alpha = 0$ seem to show a different behavior, where the area law phase is more stable to the increase in stochastic fluctuations.

## 4 Discussion

We studied the stochastic evolution of a quantum state under free unitary dynamics disrupted by the back-action of two competing weak continuous measurements. In the absence of the unitary dynamics, the steady state of the system undergoes a transition between two area-law entanglement scaling phases distinguished by the presence or absence of (symmetry-protected) topological order. We show from numerical analysis that the universal exponent of the finite size scaling differs from its analog for stroboscopic dynamics with projective measurements.

In the presence of unitary dynamics, a phase characterised by sub-volume scaling of the entanglement entropy separates the topologically distinct area-law phases. We analyze the entanglement transition along different cuts of the phase diagram using a new indicator based on a combination of half cut entanglement entropy and the topological entanglement entropy. This allows us to clearly identify a transition between area law and sub-volume scaling of the entanglement entropy at finite measurement rate along generic cuts. Conversely, we find that the entanglement transition in the commonly studied charge conserving model $\alpha = 0$ is non generic, in that our numerical data does not point to a transition at finite measurement rate.

Finally, we introduce a *partial* post-selection measurement scheme, which allows us to unambiguously connect the three phases of the weakly monitored system to those of a parent post-selected non-Hermitian Hamiltonian and, in turn, to its topological indices based on winding numbers. The connection we establish between stochastic evolution and non-Hermitian Hamiltonians sets the stage for analyzing the way in which the critical behavior is modified by the fluctuating measurement outcome, and constitutes a first step towards a topological classification of non-unitary quantum stochastic dynamics.

## Acknowledgements

We would like to thank I. Jubb, L. Coopmans, H. Schomerus, M. Szyniszeewski and Jonathan Ruhman for useful discussions. A.R. acknowledges the EPSRC via Grant No. EP/P010180/1. G.K. acknowledges the support of a DIAS Schrödinger Fellowship and the SFI Career Development Award 15/CDA/3240. D.M. acknowledges support from the Israel science foundation

(grant No. 1884/18).

# A   Physical implementation of the measurements operations

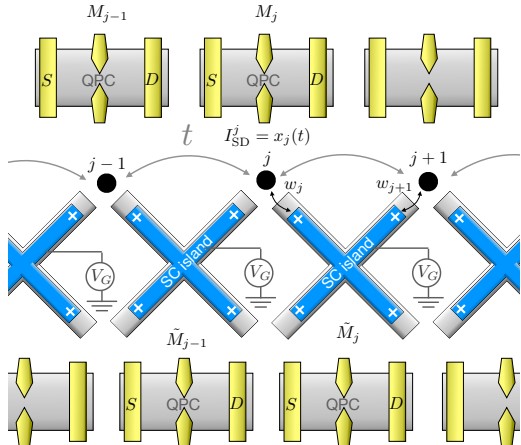

Figure 6: Detection scheme for the operator $\tilde{M}_j$ based on a Coulomb-blockade superconducting island (blue) hosting 4-Majorna zero modes (white). The fermionic chain can be realized i.e. by an array of tunnel coupled quantum dots. Two Majorana zero modes are tunnel coupled to adjacent chain sites, while the parity of the remaining two Majorana modes is read by a charge sensing detector.

We present here a possible measurement protocol to detect the operators $\tilde{M}_j$ introduced in Eq. (3). The measurement protocol is based on a Majorana-island setup. While the local density $M_j$ can be monitored via detector coupled to the system via a density-density interaction, as in commonly used charge sensors for electronic nanodevices [103, 104]), the operator $\tilde{M}_j$ requires a modification of the system's particle number. At a formal level, $\tilde{M}_j$ is a measurable operator since it is Hermitian, but it requires a careful design of the detector's coupling due to its structure in the particle-hole space.

If the system is realized as a chain of Majorana modes (e.g. at the edges of quasi-one-dimensional systems [105], the measurements of both $M_j$ and $\tilde{M}_j$ can be implemented by weak parity measurements of pairs of Majorana modes therein [106–109]. A possible detection scheme for the operator $\tilde{M}_j$ in a system where only fermionic modes (as opposed to Majorana modes) are accessible is sketched in Fig. A.

The detector, coupled to two adjacent sites $j$ and $j+1$, consists of an isolated superconducting island hosting 4 Majorana zero modes, $\eta_\alpha^j$, $\alpha = 1, ..., 4$. The superconductor is in a coulomb blockade regime where the charging energy is adjusted so that the total parity of the island is odd [110–113]. For an island hosting 4 Majorana zero modes, this results in a 2-fold degenerate space spanned by $|0, 1\rangle$ and $|1, 0\rangle$. Here $|1, 0\rangle = (\eta_1 - i\eta_2)/2 |0, 0\rangle$ and $|0, 1\rangle = (\eta_3 - i\eta_4)/2 |0, 0\rangle$ with $|0, 0\rangle$ defined by $(\eta_1 + i\eta_2) |0, 0\rangle = 0$, $(\eta_3 + i\eta_4) |0, 0\rangle = 0$. When the detector is tunnel-coupled to the system as indicated in Fig.1(b), single electron tunneling will be prohibited by Coulomb blockade, and correlated electron hopping is the only term allowed, as described by the Hamiltonian

$$H_{\text{det}} = i\tilde{t}_j \eta_1^j \eta_2^j (c_j + c_j^\dagger)(c_{j+1} - c_{j+1}^\dagger) = i\tilde{t}_j \eta_1^j \eta_2^j \tilde{M}_j \,. \tag{A.1}$$

Note that the specific combination $c_j \pm c_j^\dagger$ entering the coupling of two adjacent detectors is important, and can be adjusted by controlling the flux between adjacent detectors. We further

assume that the parity of pairs of Majorana zero modes in the detector can be continuously measured, e.g. via recently proposed protocols [106–109]. Specifically we assume to to perform a continuous readout of the the parity of the pairs of Majorana zero modes $\eta_{3,j}$ and $\eta_{4,j}$. The latter returns a continuous output $x$ with probability $P(x) = \langle\Psi|\mathcal{K}^\dagger(x)\mathcal{K}(x)|\Psi\rangle$ and an associated conditional back-action

$$|\Psi\rangle \to \frac{1}{\sqrt{P(x)}}\mathcal{K}(x)|\Psi\rangle \,, \tag{A.2}$$

where

$$\begin{aligned}
\mathcal{K} = (2\pi)^{-1/4}\big[&\exp\big(-(x+\lambda)^2/4\big)|0,1\rangle\langle 0,1| \\
+&\ \exp\big(-(x-\lambda)^2/4\big)|0,1\rangle\langle 0,1|\big]\,.
\end{aligned} \tag{A.3}$$

With this setup at hands, the proposed measurement protocol for the detection of $\tilde{M}$, consists of (i) initializing the superconducting island in a state $|\chi\rangle = [|0,1\rangle + |1,0\rangle]/\sqrt{2}$, (ii) coupling the detectors to the chain for a short time and (iii) measure the parity of the pairs of Majorana zero modes $\eta_{3,j}$ and $\eta_{4,j}$.

To show the validity of the protocol, we consider its application to the chain in a given state $|\psi\rangle$. After step (ii) above, the resulting (system-detector entangled) state is

$$\begin{aligned}
|\Psi\rangle &=\ |\psi\rangle|\chi\rangle + g\tilde{M}_j|\psi\rangle\,\eta_{i,j}\eta_{2,j}|\chi\rangle \\
&=\ \big(1 - g\tilde{M}_j\big)|\psi\rangle|0,1\rangle + \big(1 + g\tilde{M}_j\big)|\psi\rangle|1,0\rangle
\end{aligned} \tag{A.4}$$

to leading order in the system-detector coupling, where $g$ is a small parameter controlled by the strength and time-duration of the system-detector coupling. Finally, the state after (iv) is computed applying Eq. (A.2) to the state in Eq. (A.4), and the resulting state of the chain, conditional to the measurement outcome $x$ for small $\lambda$, takes the desired form of Eq. 6 with $\tilde{M}_j = (2d_jd_j^\dagger - 1)/2$. Note that the protocol is not sensitive to fine tuning of the parameters as long as the detectors is initialized in a superposition of parity states and the system-detector coupling evolution is weak.

## B Entanglement scaling transitions

We report here on our efforts to accurately determine the transition points between the two area law regions, and from those regions to the $\log L$ scaling regime. There are a number of approaches to take here. A direct route is to simply compare the half-cut entanglement entropy $\bar{S}_L$ at different lengths. For the particle number conserving limit $\alpha = 0$ this is has been studied numerically in Ref. [31], where the indication of a transition between an area law and a critical logarithmic scaling phase has been determined by fitting the scaling of $\bar{S}_L$ with the system size. In Fig. 3 (b) in the main text, we take a similar approach and plot the difference between $\bar{S}_{96}$ and $\bar{S}_{64}$ to reveal the general pattern of the phase diagram.

In order to be more precise one needs a measure that picks out the universal scaling properties of the transition. A clear and natural candidate is the topological entanglement entropy $\bar{S}_L^{\text{top}}$ (4). In the absence of unitary dynamics (i.e. with $w = 0$) this measure clearly picks out the transition point at $\gamma = \alpha$ see Fig 2. In [23], it was demonstrated that this measure also allows the transition lines from to area to volume-law. However, this does not appear to be universal. As we show in Figures 7 and 8, for large system sizes the topological entanglement entropy fails to increase monotonically in the sub-volume law region. As such, picking out crossing points at large system sizes becomes unreliable.

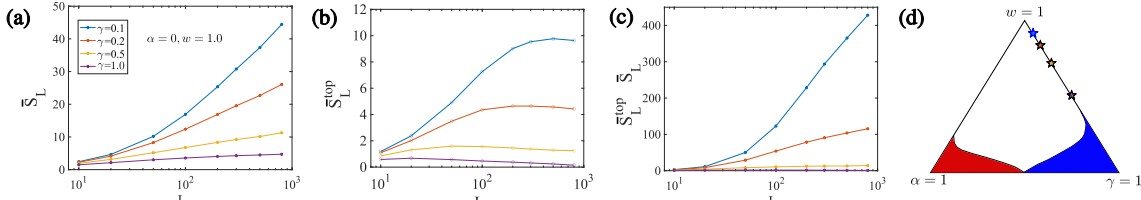

Figure 7: System size scaling of the averaged (a) half-cut entanglement entropy $\bar{S}_L$, (b) Topological entanglement entropy, $\bar{S}_L^{\text{top}}$ and (c) $\bar{S} \times \bar{S}_L^{\text{top}}$. (d) Indicates where the data points correspond to in the phase space.

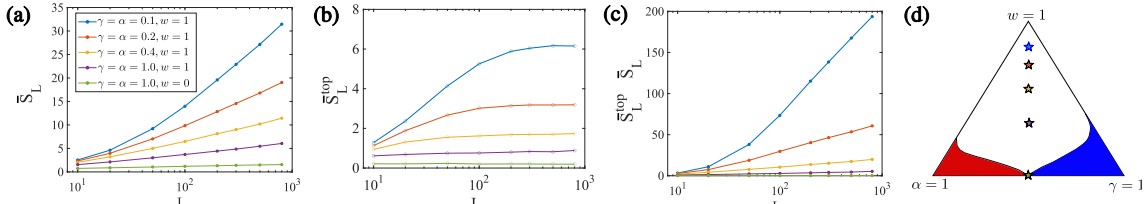

Figure 8: System size scaling of the averaged (a) half-cut entanglement entropy $\bar{S}_L$, (b) Topological entanglement entropy, $\bar{S}_L^{\text{top}}$ and (c) $\bar{S} \times \bar{S}_L^{\text{top}}$. (d) Indicates where the data points correspond to in the phase space. The special point at $w = 0$ shows log-scaling $S_L \approx 0.19 \ln L + 0.3$.

The alternative we propose here is the product of both the half-cut entanglement entropy and the topological entanglement entropy $B_L \equiv S_L^{\text{top}} \times \bar{S}_L$, calculated in the basis such that the area-law entanglement entropy tends to zero. This requirement forces the product to behave as an appropriate order-parameter in thermodynamic limit: any area-law phase will tend to zero, whereas volume or sub-volume scaling will tend to infinity.

In Fig. 3 (c) and (d) in the main text we show how this indicator behaves as it cuts across the area to $\log L$ transition line. In Figures 7 (c) and 8 (c) we also show the system size scaling behaviour in the purported $\log L$ regimes. An interesting observation is that the measure does not appear to saturate at any point along the $\gamma = \alpha$ line (vertical line through the middle of the triangle) in 8. This is in contrast to 7, where the fact that $S_L^{\text{top}}$ appears to reduce at large system sizes, leaves open the possibility that the area-law regime can include the whole $\alpha = 0$ line, excluding the $\gamma = 0$ point. We discuss this in more detail below.

## B.1 Discussion of $B_L \equiv \bar{S}_L^{\text{top}} \times \bar{S}_L$ for the particle number conserving limit $\alpha = 0$

We argued in the last section that the combination $\bar{S}_L^{\text{top}} \times \bar{S}_L$ is better suited to picking up the transition between area and sub-volume law scaling. Here we specifically wish to examine whether the crossing points of our proposed order parameter $B_L$ or $B_L'$ converge near a consistent value at different lengths $L$. To this end we examine the value $[w]_{L_a,L_b}$, defined as the $w$ along parameterized lines [here we examine $(\gamma = 0, \alpha = 1)$ (red circle), $(w = \gamma, \alpha = 1)$ (purple star),$(w = \alpha, \gamma = 1)$ (yellow star), $(w = 128\alpha, \gamma = 1)$ (green triangle), and crucially $(\alpha = 0, \gamma = 1)$ (blue circle)] such that

$$B_{L_a}(w, \alpha, \beta) - B_{L_b}(w, \alpha, \beta) = 0. \tag{B.5}$$

Away from the particle number conserving line (so with $\alpha \neq 0$) the measure $B_L$ displays consistent crossing points across a range of lengths spanning an order of magnitude, see Fig. 9 Notably this is still true if we take a phase-space cuts that are only modestly different from $\alpha = 0$ line [Cf. 9, panel (g)]. On the $\alpha = 0$ line however, the crossing point between $B_L$

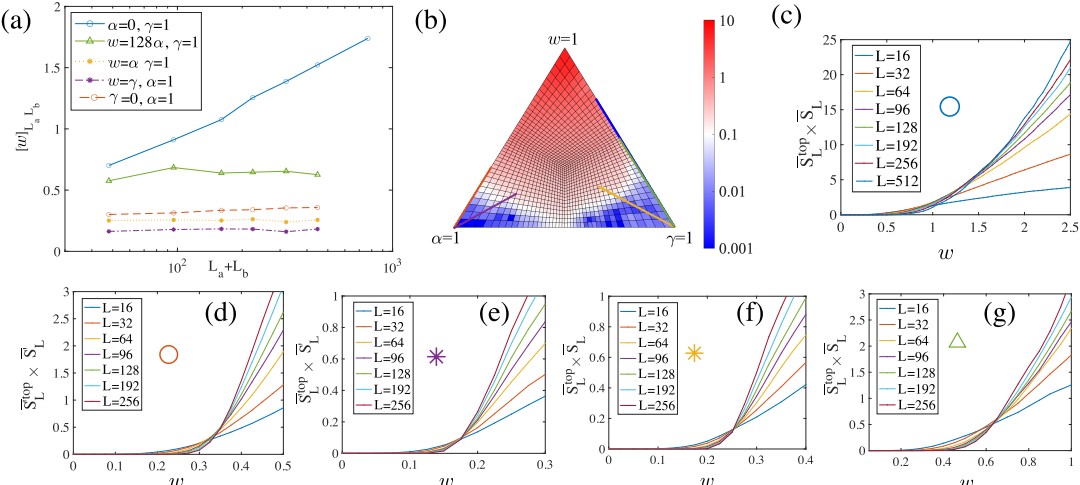

Figure 9: (a) The estimated crossing points $[w]_{L_a,L_b}$ as a function of $L_a + L_b$. (b) The phase space cuts corresponding to panels (c) through (g). Panel (c) - which corresponds with the blue circles in panel (a), and is the only data-set with particle number conservation, the crossing points between different system sizes seems to grow logarithmically with system size. This is in contrast to panels (d) through (g) which show consistent crossing points.

of different lengths scales approximately as $\log L_A + L_B$. This behavior, when extrapolated, suggests that there is no consistent crossing point when particle number is conserved.

An important question is, given the duality between $\alpha$ and $\gamma$ measurements, why are we able to find a consistent crossing point on the $\gamma = 0$ line? Here it is important to remember the way the duality mapping is achieved: the system evolves under the dual continuous monitoring terms but where the unitary dynamics is now governed by

$$H' = \frac{w}{2} \sum_i d^\dagger_{i+1} d_{i-1} + d^\dagger_{i+1} d^\dagger_{i-1} + 2 d^\dagger_i d_i - 1 + \text{h.c.} \tag{B.6}$$

and the physical cuts that we use to partition the system are made in the $d$-fermion basis. Crucially in this basis the Hamiltonian itself does *not* preserve the number of $d$-fermions.

## C  Post-selected dynamics and the corresponding non-Hermitian Hamiltonian

The topological transition in the model originates form the competing stationary states stabilized by the Zeno effects of the two measurements. In order to capture this effect, it is possible to analyze individual measurements readouts, that stabilize one of the possible measurements eigenstates. The process can be readily described in terms of detector outcome. For the case of a local density measurement, for example, it would correspond to selecting the outcome $x = a_+$ from the probability distribution in Eq. 5. The result is a deterministic evolution of the system via a non-Hermitian Hamiltonian $\mathcal{H}_\gamma = -i\gamma \sum_i M_i$. We can therefore analyze the steady state of the deterministic process determined by the overall non-Hermitian Hamiltonian

$$\mathcal{H} = H_0 - i\gamma dt \sum_{i=1}^{L} M_i - i\alpha dt \sum_{i=1}^{L-1} \tilde{M}_i \,. \tag{C.7}$$

The corresponding post selected model can be studied analytically following Ref [41]. We consider non unitary dynamics with a time evolution operator:

$$U(t) = \prod_{t=1}^{N} U_1(t)U_2(t). \tag{C.8}$$

Where the time evolution of the post selected quantum state is governed by $U_1(t) = \exp(-\tau H_{nh})$ followed by a normalization of the wave function, where

$$H_{nh} = \frac{\alpha}{2} \sum_i \left(c_i^\dagger - c_i\right)\left(c_{i+1}^\dagger + c_{i+1}\right) + \gamma \sum_i c_i^\dagger c_i \tag{C.9}$$

and the unitary dynamics is modeled by a Hermitian hopping Hamiltonian $U_2(t) = \exp(-i\tau H_h)$,

$$H_h = \sum_i w c_i^\dagger c_{i+1}, \tag{C.10}$$

the wave function dynamics is given by:

$$|\psi(T)\rangle = \frac{U(T)}{\sqrt{Z}} |\psi_0\rangle, \tag{C.11}$$

where the normalization is $Z = \langle \psi_0 | U(T)^\dagger U(T) | \psi_0 \rangle$. Under the evolution $U(T)$ the state $|\psi(T)\rangle$ remains Gaussian. Consequently the entanglement is entirely expressed through the correlation matrix:

$$\begin{pmatrix} C_{ij}(T) & F_{ij}(T) \\ -F_{ij}^*(T) & \delta_{ij} - C_{ij}(T) \end{pmatrix}, \tag{C.12}$$

$$\begin{aligned} C_{ij}(T) &= \langle \psi(T) | c_i^\dagger c_j | \psi(T) \rangle, \\ F_{ij}(T) &= \langle \psi(T) | c_i c_j | \psi(T) \rangle, \end{aligned} \tag{C.13}$$

following [41] we derive the time evolution of the correlation matrix. The resulting equations reads:

$$\begin{aligned} \frac{dC_k}{dt} &= -4\xi(k)\left[C_k(1 - C_k) + |F_k|^2\right] \\ &\quad -i2\Delta(k)(2C_k - 1)(F_k - F_k^*), \\ \frac{dF_k}{dt} &= -4\xi(k)\left[F_k(1 - 2C_k)\right] - i4w(k)F_k \\ &\quad -2i\Delta(k)(2F_k^2 + 2C_k^2 - 2C_k + 1), \end{aligned} \tag{C.14}$$

where $\xi(k) = (\alpha \cos k + \gamma)$ and $\Delta(k) = \alpha \sin k$ and $w(k) = w \cos k$. We first consider the simplified two measurement model setting $t = 0$. We seek a steady state solution. We find the following solutions:

$$\begin{aligned} F_k &= \frac{i\Delta(k)}{2\sqrt{\xi(k)^2 + \Delta(k)^2}} = \frac{i\Delta(k)}{2E(k)}, \\ C_k &= \frac{1}{2} - \frac{\xi(k)}{2\sqrt{\xi(k)^2 + \Delta(k)^2}} = \frac{E(k) - \xi(k)}{2E(k)}, \end{aligned} \tag{C.15}$$

which are the correlators in the ground state of a Kitaev chain. The entanglement entropy derived from this correlation matrix transitions between two distinct phases where the topologically non trivial phase $\gamma < \alpha$ is associated with a finite value of $S^{\text{top}} = 1$ while the trivial phase $\gamma > \alpha$ gives $S^{\text{top}} = 0$.

In the presence of a finite $w \neq 0$ the steady state solution is modified to:

$$
\begin{aligned}
F_k &= \frac{\sqrt{\rho(k)}}{2(1+\rho(k))}\left(2i\sin\phi(k)-\sqrt{\rho(k)}+\frac{1}{\sqrt{\rho(k)}}\right), \\
C_k &= \frac{1}{2}-\frac{\sqrt{\rho(k)}}{2(1+\rho(k))}\cos\phi(k),
\end{aligned}
\tag{C.16}
$$

where

$$
\rho(k) = \sqrt{\frac{\xi(k)^2+(\Delta(k)-w(k))^2}{\xi(k)^2+(\Delta(k)+w(k))^2}}, \tag{C.17}
$$

$$
\tan\phi(k) = -\frac{2\Delta(k)\xi(k)}{\xi(k)^2-\Delta(k)^2+w(k)^2}. \tag{C.18}
$$

These equations correspond to the correlators in the ground state of the Kitaev chain with non-hermitian hopping.

We therefore establish that under post selected dynamics, the quantum state evolves to the dark state of the $\mathcal{H} = H_h - iH_{nh} = -i\mathcal{K}$. Here $\mathcal{K}$ is the Hamiltonain of the Kitaev chain with non hermitian hopping, and the dark state of $\mathcal{H}$ is identified with the ground state of $\mathcal{K}$. We describe below some features of this model. The corresponding BdG Hamiltonian is given by:

$$
\begin{aligned}
\mathcal{K}_{BdG} &= \begin{pmatrix} \xi(k)+iw(k) & -i\Delta(k) \\ i\Delta(k) & -\xi(k)-it(k) \end{pmatrix} \\
&= (\xi(k)+iw(k))\tau_z + \Delta(k)\tau_y.
\end{aligned}
\tag{C.19}
$$

Besides charge conjugation symmetry this model has chiral symmetry which can be made apparent by rotating to the chiral basis

$$
\mathcal{K}_{BdG}^{\text{chiral}}(k) = \begin{pmatrix} 0 & q_1(k) \\ q_2(k) & 0 \end{pmatrix}, \tag{C.20}
$$

where

$$
q_{1,2}(k) = \xi(k)+i\left[w(k)\pm\Delta(k)\right]. \tag{C.21}
$$

In the hermitian limit $w = 0$ we have $q_1(k) = q_2(k)^*$ and the complex vector $q_1(k) = \gamma+\alpha e^{-ik}$ marks a circle of radius $\alpha$ centered around $\gamma$. In the non hermitian model the two complex vectors mark two tilted ellipses, similar to the non Hermitian SSH model [114–118]. Gapped phases of this model are dichotomized by whether the two ellipses encapsulate the origin. The transition between these two distinct behaviors occurs when the two complex vectors pass through the origin and the spectrum becomes gapless. The condition for the onset of a gapless phase is $q_{1,2}(k) = 0$ which corresponds to:

$$
\begin{aligned}
\xi(k) &= 0, \\
\Delta(k) &= \pm w(k).
\end{aligned}
\tag{C.22}
$$

Solving this set of equations, the transition to a gapless phase occurs for $\gamma = \frac{\alpha}{\sqrt{1+w^2/\alpha^2}}$. The phase diagram of the non Hermitian model exhibits three distinct phases. Two topologically distinct gapped phases, distinguished by the winding of both $\nu_{1,2} = 1$ or none $\nu_{1,2} = 0$ of the complex vectors $q_{1,2}(k)$, separated by a gapless phase, see Fig. 4.

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
