# Peer review of "Topological transitions with continuously monitored free fermions"

_SciPost Physics, doi:SciPost Phys. 14, 031 (2023)_

## Round 2 · Referee Report · Anonymous (Referee 1) · 2022-5-19

Strengths

- The paper discusses timely problems, and gives new ideas on measurement-induced phase transitions.

Weaknesses

- Some parts of the papers need clarifications.
- The writing of the manuscript can be improved and polished.

Report

## Report for the manuscript by G. Kells, D. Meidan, and A. Romito
The authors of this manuscript study a model of free fermions subject to two non-commuting continuous monitoring actions, and unitary dynamics.
By varying the control parameter of each of these actors ($\gamma$ for the $M_j$ measurements, $\alpha$ for the $\tilde{M}_j$ measurements, and $w$ for the unitary dynamics), the authors show the setup exhibits a rich entanglement and *symmetry protected* topological (SPT) phase diagram.

Alone, the competition of non-commutative measurements induces distinct SPT phases, both with area-law entanglement entropy scaling and separated by a phase transition, whose location in the parameter space is understood via duality arguments (the critical point lies exactly at the self-dual point). Adding unitary dynamics, as encoded in a fermionic hopping Hamiltonian, the phase diagram is enriched by a phase with logarithmic system size scaling (log law), and with a claimed trivial SPT phase.

These conclusions stem from numerical results, and are supported within specific regimes by analytic arguments on the non-Hermitian Hamiltonian and by analyzing partial post-selection. I also mention that some of these results are analytical, as the totally post-selected non-Hermitian Hamiltonian is quadratic in the fermionic degrees of freedom.

Lastly, the authors sketch how their setup can be implemented in actual experiments, by detailing an implementation with Coulomb blockade superconducting island.

Overall, I believe this manuscript presents relevant ideas, which tackle and clarify some open questions in the field of measurement-induced phase transitions. On the other hand, some conceptual aspects require clarification, and the presentation of the physical setup and the results is sometimes cumbersome, undermining the quality of this manuscript. (Below is the full list of required changes and of typos I found). For these reasons, at present, I cannot recommend the paper for publication in SciPost Physics. I will be happy to suggest the paper for publication once the clarifications are provided and the manuscript text is polished.

In the sections below I will provide:
1. A detailed analysis of the novelties introduced in this work, and a possible *auxiliary* improvement the authors may consider including in the future version of the manuscript
2. A list of clarification/typos/modifications required.

### The novelties of the manuscript
There are a variety of new results contained in this paper

#### 1. Continuous-time monitoring is a relevant perturbation
Symmetry-protected topological and truly topological phases have been already investigated in monitored random circuits (Ref.[14-15] of the manuscript, the paper of S. Sang and T. Hsieh arxiv:2004.09509 which is missing in the bibliography). However, this paper is the first to discuss the effect of continuous-time in systems hosting non-trivial SPT phases (to my knowledge, this is also the only one together with arxiv:2201.05341).

The first surprising result is in the measurement-only framework (no unitary part, i.e. $w=0$). Here the authors can locate exactly the critical point based on arguments on duality (the transition happens at the self-dual point), and they find a correlation length critical exponent $\nu=5/3$. This should be compared with Ref.[13], where Lang and Buchler discuss the discrete version of the measurement-only dynamics, and they as well locate the transition through duality arguments and an exact mapping to a 2D percolation problem. However, the critical exponent they find is $\nu=4/3$ (which is the percolation one: see also arxiv:2012.00031, arxiv:2108.04274, and arxiv:1911.11169).
Therefore, continuous-time measurements provide a relevant perturbation, which drives the critical point toward a new universality class.

#### 2. Partial post-selection reveals in which regimes the non-Hermitian Hamiltonian capture quantitatively the physics of the system
The non-Hermitian Hamiltonian considered emerges as the post-selection limit for the measurements $1\equiv x_j = \langle M_j\rangle + dW_j$. This object has been shown in Ref[26,36,37] to capture qualitative aspects of the measurement-induced transition. However, the clear regime of validity of this object has remained vain (for instance, the post-selected Hamiltonian does not capture the relevant physics in the specific framework of Ref.[43,47] and arxiv:2109.10837, which would be a pertinent entry to add to the bibliography).

Here the authors show that depending on the degree of post-selection performed, that is requiring that $x_j>r_c$, the physics interpolate from a regime where the non-Hermitian Hamiltonian is quantitatively describing the system $r_c\sim -1$ to a regime where the results largely influence the measurement-induced phases (e.g. $r_c\sim -2$).

*Auxiliary improvement*: within the post-selected dynamics, it has been shown in arxiv:2201.09895 that from the non-Hermitian Hamiltonian with momentum resolution one can compute analytically the scaling of the entanglement entropy within the critical region (the effective central charge). It would be interesting to see if a similar computation can be extended in the present framework.

### Clarification needed and typos
First I shall present the clarification needed, and below is a list of typos I found.

#### Clarifications needed
- In general, the authors should comment on which topological phase we are talking about. I believe for instance that the expression "*symmetry-protected*" should be stressed if this is the case (likely seems so). Indeed, topology is a broad concept in many-body quantum physics, and to call a phase truly topological one should also perform an analysis on the robustness of the phase to local perturbation, _with and without_ the symmetries of the system.
- Equation (1): the normalization can be written exactly (and results in the quantum state diffusion stochastic Schrodinger equation). I believe this would be good to spell out: indeed the normalization comes from the generalized measurement process, and is not put "by hand". For the experts in the field, this is obvious, but for the inexperienced reader, it would avoid confusion.
- Partition sizes (A,B,C,D) are not defined anywhere. I assumed they were all L/4 from the figures, but the authors should include this information somewhere.
- Last paragraph before Sec II A: the discussion about non-linear observables in the density matrix should be clarified before the introduction of the average entropies (this would smoothen the logical flow).
- BdG was never defined.
- In Eq.(10) you can exponentiate altogether the measurement operations (and not use a product of two exponential, one for each measurement protocol). When also the unitary part is present, this is irrelevant, as the bottleneck for the numerical precision is the Trotterization step. However, in the case of monitored-only dynamics, it would be the infinitesimal exact solution (in the Ito sense).
- The labels of the figures are too small: please enlarge them.
- Which is the scaling at the self-dual point $w=0, \gamma=\alpha$ of the entanglement entropy with system size? This was never shown. Furthermore, assuming the scaling is a log-law, it would be important to compare this effective central charge with the one in Ref.[13] (which is analytically obtained).
- When presenting the numerical data, please include the right errors (not $O(10^{-2})$, which is just the magnitude).
- Many $r_c<0$ in pag 9. Are these intended or are these typos?
- In Fig.4 the topological entanglement entropy is larger than 1. As I understood, in a critical system the values should be bounded to an O(1) value (cancellation of various log contributions). How can the authors rule out the topological nature of the critical phase from the numerical data?
- Regarding the experimental protocol: the authors should include also a discussion about the measurement of the topological entropy and the possible limitations due to the post-selection/partial-post-selection.
- Maybe some of the papers cited in this report should be included in a future version of the manuscript, being missed/relevant for the overall bibliography.

#### Typos:
- After Eq.(3) there is an interrupted sentence (a period before [49]).
- Broken latex in [49] ($\sigma^x$).
- last line pag 4: the operator $c_i$ kills the vacuum. There should only be some $c^\dagger_j$ operators in the definition of $|\Psi_C\rangle$.
- same but for $d_i$ and $|\Psi_K\rangle$.
- "eignestates of the topologically" in pag.5
- Double "for" in pag. 5
- After figure 4: interrupted sentence ("In particular a tri-critical point ..." there is a period before [15])
- $S'_X$ should probably be $\tilde{S}_X$
- "suffers form large"
- In Sec III C: "area-to-volume law transitions", only Gullans' reference discuss the volume to area transitions. The other references discuss area to log.
- "Fig.fig:system(b)"
- "For the case of a the "

Requested changes

1- Clarify the points raised in the manuscript
2- Correct the typos

Furthermore, I would strongly suggest to polish the writing of this manuscript in general, as it would improve the quality of the paper

  • validity: high
  • significance: high
  • originality: high
  • clarity: ok
  • formatting: below threshold
  • grammar: acceptable

Author:  Alessandro Romito  on 2022-08-25  [id 2751]

(in reply to Report 1 on 2022-05-19)
Category:
answer to question
correction

We thank the referee for the careful read of the manuscript and for the insightful suggestions to improve the presentation of our findings. We address here the points raised by the referee one by one.

REFEREE: In the sections below I will provide: 1. A detailed analysis of the novelties introduced in this work, and a possible auxiliary improvement the authors may consider including in the future version of the manuscript 2. A list of clarification/typos/modifications required.

The novelties of the manuscript

There are a variety of new results contained in this paper

1. Continuous-time monitoring is a relevant perturbation

Symmetry-protected topological and truly topological phases have been already investigated in monitored random circuits (Ref.[14-15] of the manuscript, the paper of S. Sang and T. Hsieh arxiv:2004.09509 which is missing in the bibliography). However, this paper is the first to discuss the effect of continuous-time in systems hosting non-trivial SPT phases (to my knowledge, this is also the only one together with arxiv:2201.05341). The first surprising result is in the measurement-only framework (no unitary part, i.e. w=0). Here the authors can locate exactly the critical point based on arguments on duality (the transition happens at the self-dual point), and they find a correlation length critical exponent ν=5/3. This should be compared with Ref.[13], where Lang and Buchler discuss the discrete version of the measurement-only dynamics, and they as well locate the transition through duality arguments and an exact mapping to a 2D percolation problem. However, the critical exponent they find is ν=4/3 (which is the percolation one: see also arxiv:2012.00031, arxiv:2108.04274, and arxiv:1911.11169). Therefore, continuous-time measurements provide a relevant perturbation, which drives the critical point toward a new universality class. AUTHORS: We thank the referee for appreciating the significance of this finding. We have made sure to stress this more clearly and prominently in the new version of the manuscript.

2. Partial post-selection reveals in which regimes the non-Hermitian Hamiltonian capture quantitatively the physics of the system

REFEREE: The non-Hermitian Hamiltonian considered emerges as the post-selection limit for the measurements 1≡xj=⟨Mj⟩+dWj. This object has been shown in Ref[26,36,37] to capture qualitative aspects of the measurement-induced transition. However, the clear regime of validity of this object has remained vain (for instance, the post-selected Hamiltonian does not capture the relevant physics in the specific framework of Ref.[43,47] and arxiv:2109.10837, which would be a pertinent entry to add to the bibliography). Here the authors show that depending on the degree of post-selection performed, that is requiring that xj>rc , the physics interpolate from a regime where the non-Hermitian Hamiltonian is quantitatively describing the system rc∼−1 to a regime where the results largely influence the measurement-induced phases (e.g. rc∼−2). AUTHORS: Indeed the referee is correct in reporting our finding.

REFEREE: Auxiliary improvement: within the post-selected dynamics, it has been shown in arxiv:2201.09895 that from the non-Hermitian Hamiltonian with momentum resolution one can compute analytically the scaling of the entanglement entropy within the critical region (the effective central charge). It would be interesting to see if a similar computation can be extended in the present framework. AUTHORS: The technique used in arXiv:2201.09895 can in principle be applied to our model. However, as we show in the manuscript, the steady state equations for the post-selected dynamics are solved by the correlators calculated in the ground state of the Kitaev model with non hermitian hopping. Consequently, at criticality and in the limit of vanishing unitary dynamics the correlators are given by those of the Kitaev model with central charge of c= ½. The dependence of the central charge on the non-hermiticity for the ground state of the Kitaev model is indeed interesting to explore, but we feel it is not directly within the scope of the current work whose focus is stochastic dynamics generated by competing measurement and the connection with the post-selected model.

REFEREE:

Clarification needed and typos

First I shall present the clarification needed, and below is a list of typos I found.

Clarifications needed

  • In general, the authors should comment on which topological phase we are talking about. I believe for instance that the expression "symmetry-protected" should be stressed if this is the case (likely seems so). Indeed, topology is a broad concept in many-body quantum physics, and to call a phase truly topological one should also perform an analysis on the robustness of the phase to local perturbation, with and without the symmetries of the system. AUTHORS: Indeed, the manuscript is concerned with symmetry-protected topological phases only. We have stressed this in the new version of the manuscript in section II. The degree of protection against local perturbation is an important and interesting question. Even the ground states of full (non-symmetry-protected) topological systems are known to have limited robustness to local noise, if the induced fluctuations can enable transitions to states above the gap. However, in the context of the monitored quantum systems that we are concerned with, generic states are effectively in the middle of the spectrum, and thus the gap protecting the ground state is not particularly relevant. That said, what one observes is that for sufficiently strong measurements of the SC term, almost all trajectories tend to states that have well-defined and non-trivial topological entanglement entropy. Such states are related to ground-states, but with an extensive (~approx L/2) number of fluctuating local bulk excitations. Other topological properties are also well defined. For example, for the open boundary conditions we consider, the system can also be understood to have something like a Majorana zero-mode, which connects each state to its parity-switched partner.
    A full exploration of this question is beyond the scope of what we intend here. However we have now stressed explicitly the symmetry-protected nature of the topological phases under consideration here and we have commented on the broader meaning/implications of topological protection in the context of monitored quantum systems at the end of Section IIIA and in our conclusions.

REFEREE - Equation (1): the normalization can be written exactly (and results in the quantum state diffusion stochastic Schrodinger equation). I believe this would be good to spell out: indeed the normalization comes from the generalized measurement process, and is not put "by hand". For the experts in the field, this is obvious, but for the inexperienced reader, it would avoid confusion. AUTHORS: We have provided the general form of SSE in the resubmitted version, so that the normalization is accounted for (cf. also comment from Referee 1)

REFEREE: - Partition sizes (A,B,C,D) are not defined anywhere. I assumed they were all L/4 from the figures, but the authors should include this information somewhere. AUTHORS: We refer to Fig. 1 where we now give the exact definition with partitions of sizes L/4 in the resubmitted version.

REFEREE: - Last paragraph before Sec II A: the discussion about non-linear obervables in the density matrix should be clarified before the introduction of the average entropies (this would smoothen the logical flow). AUTHORS: We have moved the discussion on averaging of non-linear observables to just before the introduction of the entropies.

REFEREE: - BdG was never defined. AUTHORS: Corrected. It is now now defined just before Eq. (8)

REFEREE: - In Eq.(10) you can exponentiate altogether the measurement operations (and not use a product of two exponential, one for each measurement protocol). When also the unitary part is present, this is irrelevant, as the bottleneck for the numerical precision is the Trotterization step. However, in the case of monitored-only dynamics, it would be the infinitesimal exact solution (in the Ito sense). AUTHORS: we agree that the proper evolution in the presence of the measurement can be included in a single exponentiation of the two operators, and we have now defined it this way before Eq. (7). Note, however, that in the numerical implementation, the process is obtained as a trotterized product of the exponents of the two measurements, because, in this way, measurements at different j commute and the exponentiation can be computed efficiently.

REFEREE: - The labels of the figures are too small: please enlarge them. AUTHORS: We have rearranged the panels in our figures and consistently adjusted the size of labels.

REFEREE: - Which is the scaling at the self-dual point w=0,γ=α of the entanglement entropy with system size? This was never shown. Furthermore, assuming the scaling is a log-law, it would be important to compare this effective central charge with the one in Ref.[13] (which is analytically obtained). AUTHORS: This is an interesting point that we have now addressed in the revised version. We find that the scaling is S_L \approx 0.19 log_e L +0.3, as reported in Fig. 8 in the newly added appendix B.

REFEREE: - When presenting the numerical data, please include the right errors (not O(10−2), which is just the magnitude). AUTHORS: We have followed the suggestion by the referee.

REFEREE: - Many rc<0 in pag 9. Are these intended or are these typos? AUTHORS: The phase diagram well approaches the post-selected one already for r_c ~0, so we have presented the values of r_c that better show visually the transition between the fully stochastic and post-selected models.

REFEREE: - In Fig.4 the topological entanglement entropy is larger than 1. As I understood, in a critical system the values should be bounded to an O(1) value (cancellation of various log contributions). How can the authors rule out the topological nature of the critical phase from the numerical data? AUTHORS: The referee is correct and we thank the referee for this question. We previously determined the nature of the phase by examining the scaling of the half-cut entanglement entropy and the topological entanglement entropy separately. As a result of this remark by the referee, we have decided to present the data combining the two quantities and plotting S^top_L x S_L. As we discuss in detail in the new draft, this combination will scale to \infty as function of L (as Log L) in the critical region, and will scale to 0 in the area law phase (with a suitably adapted S^top_L). This quantity allows us to determine the phase separation as crossing points of curves for different system sizes. We have commented extensively on this point in Section IIIB and in the new appendix B.

REFEREE: - Regarding the experimental protocol: the authors should include also a discussion about the measurement of the topological entropy and the possible limitations due to the post-selection/partial-post-selection. AUTHORS: The detection of the transition in the paper is ultimately a question on how to measure the topological entanglement entropy of the system, which in turns, poses the same challenges of detection as the half-cut entanglement entropy (or Renyi entropies). The experimental detection ultimately requires having several identical realizations of the same trajectories, so to perform a tomography of a trajectory-specific (i.e. post-selected) state. This challenging task is still open to investigation, including new theoretical proposals on how to detect the transition and promising experimental progress in small quantum circuits. — cf. Nat.Phys. 18, 760–764 (2022), arXiv:2203.04338. Analogous techniques could be adopted in principle for the detection of the topological entanglement entropy. We have added a comment about the limitations of the measurement of (topological) entanglement entropy in Section II.

REFEREE: - Maybe some of the papers cited in this report should be included in a future version of the manuscript, being missed/relevant for the overall bibliography. AUTHORS: We have included the suggested references.

REFEREE:

Typos:

AUTHORS: We have addressed all the typos

REFEREE: Requested changes 1- Clarify the points raised in the manuscript 2- Correct the typos Furthermore, I would strongly suggest to polish the writing of this manuscript in general, as it would improve the quality of the paper. AUTHORS: We have rewritten several parts of the manuscript. We are confident that the rewritten version is more pleasantly readable.

---

## Round 2 · Referee Report · Anonymous (Referee 2) · 2022-6-11

Strengths

1 - The paper is topical and timely.
2 - Introduces competing measurements in the continuous measurement framework
3 - Promotes some interesting ideas with respect to post selection and new universal behavior (however, see weaknesses, does not elaborate sufficiently on them)

Weaknesses

1 - The paper is not very accessible, one has to scroll up and down to connect different bits and pieces.
2 - The central results are not worked out very strongly and are not really elaborated on.

Report

The paper introduces a new interesting scenario, where two different, non-commuting continuous measurements are competing with each other (and later with a quadratic Hamiltonian). The paper, although interesting, is, however, not strong enough for SciPost Physics since it only scratches the surface of the potential (!) new phenomena observed in this setting of free fermions. Given the simplicity of the latter and the bulk of work available on free fermions subject to projective and continuous measurements already, I do not recommend publication in SciPost Physics but SciPost Physics Core instead.

Reasons that lead to my conclusion:
(a) A new universality class for topological phase transitions due to continuous measurements is proposed. However, only one critical exponent for rather small system sizes is extracted and it is not further discussed. Why should we expect a different exponent at all for continuous measurements compared to stroboscopic measurements? Does this contradict earlier works from some of the authors that showed that stroboscopic measurements and continuous measurements yield the same universal behavior? On a coarse grained level, I see no reason to expect a different phenological explanation for this area-to-area law transition than what is found in exactly the same type of setup but for stroboscopic measurements (arXiv:2108.04274, Ref [13]). What explanation do the authors have for different critical behavior?

(b) The phase transition between the area law phases and the extensive/subextensive regimes is barely discussed (the true transition, not the non-hermitian one). This analysis is in my opinion not really contributing any substantial information to the paper. It is rather presenting results on a more phenomenological level and confirms what one would expected from previous work [26,45]. Reading it felt more distracting than enlightening.

(c) For the non-hermitian postselected evolution: if one considers only the measurement terms then the system evolves into the ground state of a Hermitian Hamiltonian, which is just the Ising model in the traverse field. The transition line agrees of course completely with the one obtained by measurements (which it must due to the duality) but the phase transition itself is simply the ground state phase transition in the Ising model in a transverse field. Since the critical behavior of this model is fundamentally different from the true measurement-only model, I don't see a reason why one should believe any of the predictions in the extended critical regimes (for H_0 nonzero). Then a large part of the phase diagram displays extensive or subextensive entanglement scaling where I expect the differences to be even more dramatic.

I believe analysing the non-Hermitian model can yield insights on certain types of trajectories (where always the same type of measurement result is observed and does not fluctuate in time). However, the true dynamics of measurement seem to be strongly modified by the fluctuating measurement outcomes, yielding predictions from postselected trajectories qualitative at best. This separation is not clearly worked out in the paper and one gets the impression that the postselected trajectories explain the measurement dynamics sufficiently well (apart from minor details). I think the opposite is true.

Comments:
1) The simplified form of the SSE in Eq. (1) is problematic. For general measurement operators M_j, the absence of the expectation values in the noise part of the Hamiltonian exclude the presence of a well-defined measurement-dark state. I believe that for the measurements considered in this paper, this turns out to not be a problem since both correspond to densities of non-conserved quantities. In general, however, the absence of such a dark state leads to a different stationary state, as has been found in Refs. [34,35]. In order to avoid confusion of general readers, I would put the full, correct quantum state diffusion equation in Eq. (1).

2) For the detector outcomes x=<M>+W it is unclear to me what W is in those equations. The quantum state diffusion protocols have a well-defined outcome, which relates to the noise in the SSE.

  • validity: good
  • significance: good
  • originality: high
  • clarity: ok
  • formatting: excellent
  • grammar: excellent

Author:  Alessandro Romito  on 2022-08-25  [id 2752]

(in reply to Report 2 on 2022-06-11)

We thank the referee for the detailed feedback and critiques. The referee indicates that the paper is “topical and timely”. That it “Introduces a new interesting scenario of competing continuous measurements”, and “promotes interesting ideas such as a new universal behavior, and relation to a post selected model”. However, the referee lists as weaknesses the fact that the paper is not very accessible, and that the central results are not elaborated on, namely the paper only “scratches the surface” of the potential of the ideas introduced in the manuscript. We discuss these general complaints below along with other more specific questions/recommendations. We have also stressed the significance of our results in the resubmission letter .

REFEREE: The paper introduces a new interesting scenario, where two different, non-commuting continuous measurements are competing with each other (and later with a quadratic Hamiltonian). The paper, although interesting, is, however, not strong enough for SciPost Physics since it only scratches the surface of the potential (!) new phenomena observed in this setting of free fermions. Given the simplicity of the latter and the bulk of work available on free fermions subject to projective and continuous measurements already, I do not recommend publication in SciPost Physics but SciPost Physics Core instead. AUTHORS: We are a bit perplexed by the referee’s argument that we have only scratched the surface of this phenomena in the free-fermion setting. We ourselves have remarked on the potential of the partial post selection idea, which we introduced, in linking a multitude of topological fermionic systems with continuously monitored quantum circuits. This is one of the important contributions of our work - no one else has made this connection. It does not seem reasonable to argue that our work is somehow dilute because there is so much yet to do in this arena. The referee also suggests that our work is less impactful because there are other works already available on free fermions subjected to continuous measurement. As far as we are aware, we have already cited all of those relevant works but we note that (1) None of the previous works are applicable to the wider free-fermion setting that includes superconductivity. This is crucial because we present evidence that the entanglement transition in a particle conserving model might be non generic. (2) None of those free-fermion works have studied this type of continuous dual-measurement approach in a weak measurement setting. (3) None of those works have introduced anything like our partial post-selected idea.
It is therefore clear that we have gone well beyond what can otherwise be found in the literature. Adding to this the recognized potential for further research in this realm only reinforces the case for SciPost Physics.

REFEREE: Reasons that lead to my conclusion: (a) A new universality class for topological phase transitions due to continuous measurements is proposed. However, only one critical exponent for rather small system sizes is extracted and it is not further discussed. Why should we expect a different exponent at all for continuous measurements compared to stroboscopic measurements? Does this contradict earlier works from some of the authors that showed that stroboscopic measurements and continuous measurements yield the same universal behavior? On a coarse grained level, I see no reason to expect a different phenological explanation for this area-to-area law transition than what is found in exactly the same type of setup but for stroboscopic measurements (arXiv:2108.04274, Ref [13]). What explanation do the authors have for different critical behavior? AUTHORS: Weak continuous measurements differ from stroboscopic projective measurements in two aspects: (1) In each time-step, the back action of the measurement on the state results in a nonlinear change that depends on the state of the system itself , and (2) In the continuum limit, the time step and the measurement strength both go to zero such that their ratio remains fixed. Crucially, weak measurements are non-projective; hence they do not disentangle the measured degree of freedom but rather only weaken the entanglement to a degree set by the state itself. Different universal exponents of the entanglement transitions for weak vs. projective measurements were reported in Ref. [11(arXiv:1903.05452), 25(arXiv:2005.01863)] for interacting models. (Yet numerical results indicate that the finite size scaling is the same for continuous vs. stroboscopic measurements). Therefore, it appears that generic POVM vs projective nature of the measurement is key in discriminating the phase transition class. Moreover, measurement-only models with stroboscopic projective measurements (as in arXiv:2108.04274, Ref[13] et al.) can be mapped onto a 2D classical percolation model, and the result is confirmed numerically. The projective nature of the measurement is essential in the mapping and the argument cannot be directly applied to generic POVM backaction. Based on these results, it is reasonable to expect that the universality class of weak measurements will differ from projective ones. However, we agree that follow-up work is needed to establish an analytic mapping, which goes beyond simple percolation, for the weak measurements scenario. We have largely rewritten the discussion of the results in section IIIA of the manuscript to clarify the picture and present more prominently the significance of the results.

REFEREE: (b) The phase transition between the area law phases and the extensive/subextensive regimes is barely discussed (the true transition, not the non-hermitian one). This analysis is in my opinion not really contributing any substantial information to the paper. It is rather presenting results on a more phenomenological level and confirms what one would expected from previous work [26,45]. Reading it felt more distracting than enlightening. AUTHORS: The analysis of the full model is discussed, first of all, to present a full phase diagram. While we agree with the referee that the general features of the phase diagram are to be expected, a couple of important points were glossed over: It is a significant challenge to pick out the precise boundary between subvolume and area law scaling for a free fermion model. In the new version we elaborate on this point and suggest the combination of the topological entanglement entropy and the half-cut entanglement entropy as a better indicator to determine the transition point.
Importantly, our analysis of the full phase diagram indicates that the transition between area law and subvolume law for the free, particle-conserving model studied in [Ref. 45 (arXiv:2005.09722)] is non-generic. Using our new identifier, we can determine the transition into volume law scaling at finite measurement rate, when particle number conservation is broken (e.g. in the presence of the superconducting-like measurement or in the unitary dynamics for the d-fermions). Interestingly, our numerical data cannot identify a transition at finite measurement rate for the particle conserving model. We note that previous works that study the particle conserving model could only estimate the transition based on the logarithmic fitting of the half-cut entanglement entropy.
We have now explicitly presented our data using the combination of S^top_L and S_L and stressed the new findings emerging from this analysis.

REFEREE: (c) For the non-hermitian postselected evolution: if one considers only the measurement terms then the system evolves into the ground state of a Hermitian Hamiltonian, which is just the Ising model in the traverse field. The transition line agrees of course completely with the one obtained by measurements (which it must due to the duality) but the phase transition itself is simply the ground state phase transition in the Ising model in a transverse field. Since the critical behavior of this model is fundamentally different from the true measurement-only model, I don't see a reason why one should believe any of the predictions in the extended critical regimes (for H_0 nonzero). Then a large part of the phase diagram displays extensive or subextensive entanglement scaling where I expect the differences to be even more dramatic. I believe analysing the non-Hermitian model can yield insights on certain types of trajectories (where always the same type of measurement result is observed and does not fluctuate in time). However, the true dynamics of measurement seem to be strongly modified by the fluctuating measurement outcomes, yielding predictions from postselected trajectories qualitative at best. This separation is not clearly worked out in the paper and one gets the impression that the postselected trajectories explain the measurement dynamics sufficiently well (apart from minor details). I think the opposite is true. AUTHORS: We completely agree with the referee assessment that post-selected trajectories do not explain the full measurement dynamics, quite the opposite. We thank the referee for pointing out this unintentional misunderstanding which has been clarified in the corrected version of the manuscript, in particular in section IIIC. Nonetheless, the post-selected model is valuable as it presents distinct topological and gapless phases. One of the main contributions of the present work is the observation that the true stochastic dynamics and the post-selected model are both limiting behaviors of one parent model with partial post selection. The introduction of this partial post-selected model establishes a platform to study (numerically and hopefully analytically) the relation between these two limiting behaviors. This poses some interesting challenges for future studies. For example one can now ask how the post-selected transition is modified at different r_c? Similarly we can ask if the universality class is modified for arbitrarily small deviations from the true stochastic dynamics?” These are questions that we intend to address in future works. We have now stressed the implications of our results in Section IIIC and in the conclusions.

REFEREE: Comments: 1) The simplified form of the SSE in Eq. (1) is problematic. For general measurement operators M_j, the absence of the expectation values in the noise part of the Hamiltonian exclude the presence of a well-defined measurement-dark state. I believe that for the measurements considered in this paper, this turns out to not be a problem since both correspond to densities of non-conserved quantities. In general, however, the absence of such a dark state leads to a different stationary state, as has been found in Refs. [34,35]. In order to avoid confusion of general readers, I would put the full, correct quantum state diffusion equation in Eq. (1). AUTHORS: We agree with the referee that this might generate confusion, so we have presented the full equation to start with. We have therefore replaced Eq. (1) with the general form of the SSE.

REFEREE: 2) For the detector outcomes x=<M>+W it is unclear to me what W is in those equations. The quantum state diffusion protocols have a well-defined outcome, which relates to the noise in the SSE. AUTHORS: The x_j=<M_j>+\delta W_j (apologies for the typo) is introduced just after the SSE in Eq. 1, where we point to appendix 1 for a full explanation of its meaning. In a nutshell, we regard the SSE as derived from a quantum detection model involving a linearly coupled pointer with coordinate x_j and conjugate variable p_j. Now, x_j is the outcome of the projective measurements on the pointer (cf. Appendix 1 for details). The post-selected values defined by x_j=<M_j>+\delta W_j=1 correspond to the most-likely outcome of the pointer coordinate if the system is in the eigenstate of the occupation number n_j=1 of the j-th site of the chain. We have fixed the typos and pointed out explicitly our definition of x_j in the text and pointed to the more detailed description in the appendix

---

## Round 3 · Referee Report · Anonymous (Referee 1) · 2022-8-26

Report

The authors have successfully addressed the main criticisms I presented in my past report. In addition, they have improved the manuscript readability, corrected previous typos and misleading sentences, and improved the formatting of the figures.

For these reasons, I believe the manuscript is now suitable for publication in Scipost Physics.

---

## Round 3 · Referee Report · Anonymous (Referee 2) · 2022-10-8

Strengths

  • studies new type of competition between two non-commuting, continuous measurement protocols
  • the idea of non-ideal post-selection adds a new twist to the question of experimental realisations of measurement-induced dynamics
  • a newly identified critical behavior, though not yet fully resolved, fuels the search for universality classes beyond the previously discovered ones in free fermion and unitary circuit models

Report

I acknowledge the authors' significant effort to rearrange and redo the manuscript. It is now very clear and very well structured. The results and their discussion are very accessible. All my previous criticism has been overcome.

Given the total package of strengths outlined above and the timeliness of the topic. I can fully recommend publication of the manuscript in SciPost physics.

---

## Round 3 · Author Response

Dear Editor,

We thank the referees for their comments and critiques. We feel that addressing these issues has greatly improved the manuscript.

Both referees acknowledge that the manuscript deals with an interesting and timely topic and recognize the importance of the new ideas and perspectives presented in the manuscript. However, both also comment on some shortcomings of the presentation. Indeed, one one referee comments that our central results are not elaborated on, and that the paper only “scratches the surface” of the potential of the introduced ideas.

While we acknowledge some shortcomings of the presentation, and we deal with these comments individually in the responses to the Referees' reports, we feel our work goes beyond scratching the surface and incorporates some new and innovative features that will yield many more fruitful results. Moreover, the potential for further research in no way minimizes the impact of this work. Indeed we think this is why the manuscript meets the criteria: “Open a new pathway in an existing or a new research direction, with clear potential for multipronged follow-up work”. Of course, we could always expand upon every aspect of our work. However, we feel that given the many avenues open to us from here, it is important that we first present these ideas and key initial findings as concisely and accurately as possible.

In the new iteration of the manuscript we have taken on-board the suggestions and criticisms to better highlight the main messages of the manuscript. For clarity we stress here the significance of our findings. Our work shows for the first time that competing continuous non-projective measurements in lattice fermionic systems can drive a topological entanglement transition between two area-law phases. In particular:

  1. We show that weak measurements drive a topological transition of a different universality class compared to projective measurements. This follows from the different critical behavior of the two models (see our detailed comment to referee 1 below).

  2. In the full (unitary + measurement) dynamical evolution, the model demonstrates an extended critical phase with a logarithmic entanglement scaling. In the new manuscript we show that the combination of S^top_L and S_L more clearly marks this critical scaling transition.

  3. Our analysis shows that the entanglement transition in the previously studied charge conserving model [Ref 45 (arXiv:2005.09722 )] ($\alpha= 0$ line) is non generic. In particular our numerical data does not point to a transition at finite measurement rate.

  4. We introduce a simple-to-apply model of partial post-selection that recovers the full post-selected dynamics (for which we can map the full phase diagram and associated topological indices) as a continuum limit of a family of models with stochastic dynamics. This allows one to interpolate between the fully stochastic dynamics and the post-selected models. We feel this approach will be invaluable in future studies.

We are confident that the new version can be published in Scipost Physics.

Sincerely,

Graham Kells, Dganit Meidan, Alessandro Romito

---

## Round 3 · List of Changes

• rewritten part of the abstract and changed part of the introduction to clarify the significance of our findings and polish the writing
  • rephrased part of section II
  • added paragraphs in section III to improve clarity
  • added paragraphs to section IIIA to expand our comments on the results therein
  • rewritten section IIIB to present in a new way the combined role of the entanglement entropy and topological entanglement entropy
  • rephrased parts of section IIIC to clarify the role of the partial post-selection and and post-selection models
  • rewritten part of the conclusions
  • added new appendix B
  • replaced Eq. (1) more more general formulation -changed Eq. (7) for clarity
  • rearranged Fig. 2
  • changed Fig. 3 (c-d) to present the combined results of topological entanglement entropy and half-cut entanglement entropy
  • added new Fig. 8 and Fig. 9 in Appendix B
  • corrected typos
  • added/updated references

---

## Editorial Decision

published